# Resolving Blind Inverse Problems under Dynamic Range Compression via Structured Forward Operator Modeling

**Muyu Liu** [1]  **Xuanyu Tian** [1]  **Chenhe Du** [1]  **Qing Wu** [1]  **Hongjiang Wei** [2]  **Yuyao Zhang** [1]

## Abstract

Recovering radiometric fidelity from unknown dynamic range compression (UDRC), such as low-light enhancement and HDR reconstruction, is a challenging blind inverse problem, due to the unknown forward model and irreversible information loss introduced by compression. To address this challenge, we first identify monotonicity as the fundamental physical invariant shared across UDRC tasks. Leveraging this insight, we introduce the **cascaded monotonic Bernstein** (CaMB) operator to parameterize the unknown forward model. CaMB enforces monotonicity as a hard architectural inductive bias, constraining optimization to physically consistent mappings and enabling robust and stable operator estimation. We further integrate CaMB with a plug-and-play diffusion framework, proposing **CaMB-Diff**. Within this framework, the diffusion model serves as a powerful geometric prior for structural and semantic recovery, while CaMB explicitly models and corrects radiometric distortions through a physically grounded forward operator. Extensive experiments on a variety of zero-shot UDRC tasks, including low-light enhancement, low-field MRI enhancement, and HDR reconstruction, demonstrate that CaMB-Diff significantly outperforms state-of-the-art zero-shot baselines in terms of both signal fidelity and physical consistency. Moreover, we empirically validate the effectiveness of the proposed CaMB parameterization in accurately modeling the unknown forward operator.

---

[1]ShanghaiTech University [2]Shanghai Jiao Tong University. Correspondence to: Yuyao Zhang <zhangyy8@shanghaitech.edu.cn>.

*Proceedings of the 43rd International Conference on Machine Learning*, Seoul, South Korea. PMLR 306, 2026. Copyright 2026 by the author(s).

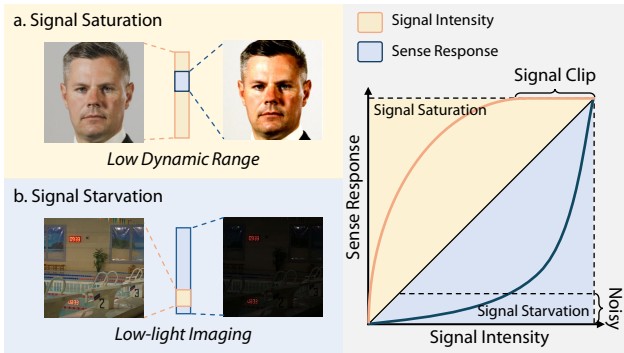

*Figure 1.* Illustration of **unknown dynamic range compression** (UDRC), which arises from a mismatch between the signal span (yellow) and the sensor's effective response region (blue).

## 1. Introduction

In digital imaging, limitations in sensing hardware and acquisition conditions hinder the faithful capture of the full radiometric dynamic range of the physical world. This pervasive issue, known as **unknown dynamic range compression** (UDRC), stems from a structural mismatch between the intrinsic signal span and the sensor's effective response (Debevec & Malik, 2023; Grossberg & Nayar, 2003; Campbell-Washburn et al., 2019). This mismatch results in dual-mode information loss: signal saturation, where high-intensity details exceed sensor capacity (*e.g.*, clipping in HDR), and signal starvation, where weak signals are compressed into the noise floor (*e.g.*, photon scarcity in low-light imaging), as illustrated in Fig. 1 (Eilertsen et al., 2017; Chen et al., 2018).

Resolving UDRC is mathematically formulated as a non-linear inverse problem with an unknown, form-agnostic forward operator. Existing zero-shot solvers struggle to navigate a fundamental bias-variance dilemma (Fei et al., 2023; Gou et al., 2024). On one end, simplified parametric models exhibit high bias. For instance, GDP (Fei et al., 2023) reduces the operator to a linear affine transformation, which is insufficient to model complex radiometric curves and often suffers from convergence instability. On the other end, unconstrained deep neural adapters exhibit high variance. Methods like TAO (Gou et al., 2024) em-

ploy Convolution Neural Networks (CNNs) to approximate the operator. Despite adversarial constraints, these over-parameterized networks are prone to overfitting measurement rather than learning the authentic degradation physics. While task-specific heuristics exist—such as manual exposure priors in low-light enhancement (Lin et al., 2025; Lv et al., 2024) or contrast enhancement in low-field MRI (Lin et al., 2024)—they often lack transferability across diverse DRC regimes or yield suboptimal solutions.

In recent years, diffusion probabilistic models (DPMs) (Ho et al., 2020; Song et al., 2020a) have emerged as a powerful paradigm for solving ill-posed inverse problems (Chung et al., 2022; Mardani et al., 2023; Zhang et al., 2025). However, standard solvers typically rely on a known forward operator (e.g., super-resolution) or a predefined analytical form (e.g., blind deconvolution) (Chung et al., 2023; Murata et al., 2023).

In the context of UDRC, the distinct challenge lies in estimating the forward operator *from scratch* concurrently with signal recovery. If the learned operator is structurally flawed (e.g., high bias) or optimization-unstable (e.g., high variance), it injects systematic errors into the measurement consistency guidance. This results in erroneous gradient fields that misdirect the reverse diffusion trajectory, causing the generative process to fail in recovering the true radiometric fidelity despite the strength of the prior.

To navigate this treacherous optimization landscape without paired supervision, we must ground the operator learning in a fundamental invariant that holds across domains. Despite the disparate physical origins of UDRC—ranging from Bloch relaxation dynamics in magnetic resonance imaging (MRI) (Rooney et al., 2007; Campbell-Washburn et al., 2019) to non-linear photoelectric responses in computational photography (Grossberg & Nayar, 2003)—these processes share a unifying physical law: the latent signal is projected through an unknown, yet *monotonic* mapping. Crucially, while this mapping distorts the metric distance between intensities, it rigorously preserves their **topological ordinality** (Sill, 1997; Gupta et al., 2016). This property provides a robust "physical anchor" enabling us to constrain the search space of the forward operator to a physically meaningful manifold. By enforcing this hard constraint, we effectively eliminate the high-variance search directions that lead to the aforementioned optimization instability, ensuring that the learned operator respects the causality of signal intensity while maintaining generalization capabilities.

To mathematically instantiate this physical constraint, we propose a unified framework, **CaMB-Diff**, grounded in **structured forward operator modeling**. We move beyond generic neural approximations by embedding the monotonicity directly into the network topology via the proposed **cascaded monotonic Bernstein (CaMB)** operator. Unlike

unconstrained networks, CaMB imposes a **hard architectural inductive bias** that strictly confines the hypothesis space to the manifold of physically consistent mappings *by construction* (Lorentz, 2012). This design effectively eliminates the learning of spurious spatial correlations (e.g., noise fitting) while retaining the high-order expressivity required to model intricate response curves. By integrating this differentiable operator into a pre-trained diffusion prior, our framework achieves **structural disentanglement**: the diffusion model recovers missing geometric details (e.g., clipped highlights), while the CaMB operator explicitly calibrates global radiometric consistency.

Our contributions are summarized as follows:

1. **Unified Perspective:** We propose a novel framework that unifies disparate signal degradation problems (Low-light, HDR, Low-field MRI) under the formalism of unknown dynamic range compression (UDRC). Crucially, we identify *monotonicity* as the fundamental physical invariant that bridges these distinct imaging modalities.

2. **Structured Operator Design:** We propose CaMB, which embeds physical monotonicity directly into architecture via cascaded Bernstein polynomials, resolving the bias-variance dilemma in operator estimation.

3. **SOTA Performance:** Extensive experiments on Low-Light Enhancement, Low-Field MRI, and HDR Reconstruction demonstrate that CaMB-Diff achieves state-of-the-art performance among zero-shot solvers and effectively competes with task-specific specialists.

## 2. Background

### 2.1. Solving Inverse Problems

In digital imaging, the acquisition process is commonly modeled as:

$$\boldsymbol{y} = \boldsymbol{A_\theta}(\boldsymbol{x}) + \boldsymbol{n}, \tag{1}$$

where $\boldsymbol{y}$ is the measurement, $\boldsymbol{x}$ is the desired signal, $\boldsymbol{n}$ is the noise and $\boldsymbol{A_\theta}$ denotes the forward operator parameterized by $\boldsymbol{\theta}$. In many classical inverse problems, the forward model admits an explicit or well-approximated analytic form, such as a convolution with a known blur kernel in non-blind image deblurring. Thus, recovering $\boldsymbol{x}$ is commonly formulated as a regularized optimization problem:

$$\hat{\boldsymbol{x}} = \arg\min_{\boldsymbol{x}} \|\boldsymbol{A_\theta}(\boldsymbol{x}) - \boldsymbol{y}\|_2^2 + \lambda \mathcal{R}(\boldsymbol{x}), \tag{2}$$

where the first term enforces data fidelity and $\mathcal{R}(\boldsymbol{x})$ reflects prior knowledge about the desired signal. However, under UDRC, the acquisition process is fundamentally ambiguous and signal-dependent, causing an unknown analytic forward

operator formulation. Consequently, approximating or implicitly learning the forward operator is essential for faithful signal recovery.

## 2.2. Operator Modeling for UDRC

Existing methods for modeling the unknown forward operator $A_\theta$ in UDRC can be categorized into *simplified parametric approximation* and *neural operator modeling*.

For the parametric approximation, early methods typically adopt histogram equalization and gamma correction, relying on simple global statistics or rigid analytical forms. More recently, GDP (Fei et al., 2023) extend the operator as a spatially varying linear affine transformation (*i.e.*, $y = \alpha x + \beta$, where $\alpha \in \mathbb{R}, \beta \in \mathbb{R}^d$). However, these methods remain fundamentally constrained by simplified or linear model assumptions, limiting their ability to capture the complex non-linear sensor responses commonly observed in real-world imaging systems and often resulting in suboptimal restoration performance.

To bypass the rigidity of parametric models, recent approaches utilize deep neural networks to approximate the forward operator. For instance, TAO (Gou et al., 2024) optimizes a CNN-based adapter during inference under adversarial guidance. However, optimizing such high-dimensional networks in zero-shot setting is highly ill-posed. Without explicit constraints, the optimization is prone to overfitting the measurement rather than learning the underlying physics, which introduces intensity inversions or structural artifacts.

## 2.3. Diffusion Models for Blind Inverse Problems

Diffusion models (Ho et al., 2020; Song et al., 2020b) with their strong capacity to model complex data distributions have demonstrated remarkable effectiveness in solving inverse problems (Chung et al., 2022; Zhu et al., 2023; Du et al., 2024). Most existing diffusion-based inverse problem solvers assume a known forward degradation model with a predefined analytical form. Even in blind inverse problem settings (Chung et al., 2023; Murata et al., 2023), the blindness typically refers to unknown parameters within fixed analytical framework(*i.e.*, unknown $\theta$ in $A_\theta$). Crucially, resolving these problems relies heavily on imposing strong priors on the unknown parameters (*e.g.*, kernel sparsity or smoothness) to constrain the solution space and mitigate the ill-posedness of the joint estimation.

In contrast, UDRC presents a more severe challenge of *functional agnosticism*, where the forward operator lacks a known explicit form. This absence of structural knowledge significantly exacerbates the ill-posed nature of the problem. Without a physical anchor, attempting to approximate the operator necessitates a perilous compromise: employing simplified parametric models introduces high bias, while

utilizing unconstrained neural networks incurs high variance. In this regime, the resulting inaccurate or unstable operator estimation injects erroneous guidance gradients into the diffusion process, which easily override the generative prior, leading to convergence failures or hallucinated artifacts that violate physical consistency.

## 3. Preliminaries

### 3.1. Forward Model of UDRC

In unknown dynamic range compression (UDRC) tasks, a complex and unknown forward operator $\mathcal{M} : \mathbb{R}^d \to \mathbb{R}^d$ maps an ideal signal $x \in \mathbb{R}^d$ to a noisy observation $y \in \mathbb{R}^d$, expressed as:

$$y = \mathcal{M}(x) + n, \quad n \sim \mathcal{N}(0, \sigma^2 I). \tag{3}$$

Learning an accurate $\mathcal{M}$ from the full function space $\mathbb{R}^d \to \mathbb{R}^d$ from a single observation is mathematically ill-posed. Therefore, rather than solving $\mathcal{M}$ in an unconstrained manner, we restrict the solution space by analyzing and exploiting the physical characteristics of UDRC based on radiometric response functions.

### 3.2. Analysis Characteristics of UDRC

Following prior analysis of camera response functions (Grossberg & Nayar, 2003), we observe that UDRC differs fundamentally from spatial degradations (*e.g.*, blur), as it does not mix geometric information but instead operates on signal intensities.

**Assumption 3.1.** $\mathcal{M}$ is *spatially stationary and univariate*, *i.e.*, the response at each location depends only on the local signal intensity and is independent of spatial position.

**Corollary 3.2.** *If $\mathcal{M}$ is spatially stationary and univariate, $\mathcal{M}$ can be expressed as a point-wise scalar function applied element-wise:*

$$[\mathcal{M}(x)]_i = m(x_i), \tag{4}$$

*where $m : \mathbb{R} \to \mathbb{R}$ is a scalar mapping.*

**Assumption 3.3.** The mapping $m(\cdot)$ is *monotonic*, *i.e.*, for any two signals $u, v \in \mathbb{R}$, if $u \leq v$, then $m(u) \leq m(v)$.

### 3.3. Hypothesis Space of UDRC Forward Model

Based on two physically motivated assumptions, we can restrict the search space of the forward operator to a physically consistent hypothesis space, formalized as:

$$\mathcal{H}_{\text{mono}} \triangleq \{\mathcal{M} \mid [\mathcal{M}(x)]_i = m(x_i), m \in C^0(\mathbb{R}), \\ m \text{ is monotonic increasing}\} \tag{5}$$

By constraining the operator in $\mathcal{H}_{\text{mono}}$, the blind estimation of $\mathcal{M}$ is transformed into a structured manifold optimization, ensuring that any learned solution respects the physical causality of signal intensities.

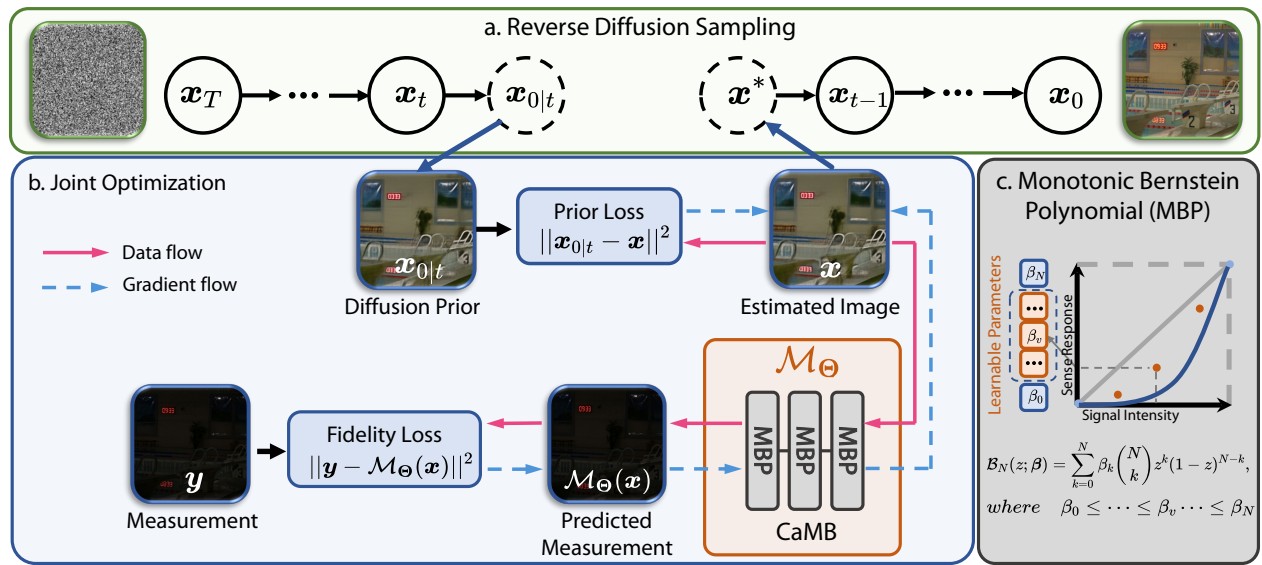

*Figure 2.* **Overview of the CaMB-Diff**: We solve unknown dynamic range compression (UDRC) via joint zero-shot optimization within the reverse diffusion sampling process. **a.** The diffusion model progressively refines the latent image through reverse-time sampling. **b.** At each sampling step, the estimated image is passed through the CaMB-parameterized forward model $\mathcal{M}_\Theta$ to produce a predicted measurement, which is matched to the observed measurement via a data-fidelity loss, while the image is simultaneously regularized by the diffusion prior through a prior loss. **c.** CaMB is constructed by cascading monotonic Bernstein polynomials, enforcing physical monotonicity by design and enabling robust estimation of the unknown radiometric mapping.

# 4. Proposed Method

Our goal is to restore a high-quality signal from a single noisy compressed measurement (*i.e.*, zero-shot setting). To achieve this, we propose using **cascaded monotonic Bernstein (CaMB)** polynomial to parameterize the unknown forward operator $\mathcal{M}$. Leveraging the physically constrained CaMB operator with the generative prior of diffusion models, we jointly optimize the recovered image and the parameterized forward model. Fig. 2 illustrates the workflow.

## 4.1. Parameterizing Forward Model via CaMB

We adopt the distinct property of the Bernstein polynomial to construct a monotonic function. In addition to enhancing expressive power while maintaining efficient optimization, we introduce a cascaded design that composes multiple low-degree Bernstein polynomials to parameterize the hypothesis space $\mathcal{H}_{\mathrm{mono}}$ defined in Eq. 5.

**Monotonic Bernstein Polynomial.** A Bernstein polynomial of degree $N$ maps an input $z \in [0,1]$ as:

$$\mathcal{B}_N(z;\boldsymbol{\beta}) = \sum_{k=0}^{N} \beta_k \binom{N}{k} z^k (1-z)^{N-k}. \quad (6)$$

By constraining the a non-decreasing order on the coefficients, *i.e.*, $\beta_0 \leq \cdots \leq \beta_N$, the polynomial is guaranteed to be monotonic on $[0,1]$, since *i.e.*, $\frac{d}{dz}\mathcal{B}_N \geq 0$. As a result,

the functional monotonicity requirement can be reduced to a simple algebraic ordering constraint on the parameters.

**Ensuring Monotonicity via Explicit Constraints.** To obtain the required non-decreasing coefficient sequence $\boldsymbol{\beta} \triangleq \{\beta_i\}_{i=0}^{N}$, we adopt a reparameterization strategy based on the cumulative sum of a learnable Softmax distribution, which enforces the monotonicity of $\boldsymbol{\beta}$ by construction. Specifically, we introduce unconstrained parameters $\boldsymbol{w} \in \mathbb{R}^N$ and map them through a Softmax function to obtain $\boldsymbol{p} = \texttt{Softmax}(\boldsymbol{w})$. The Bernstein coefficients are then defined as:

$$\beta_0 = 0, \quad \beta_k = \sum_{j=1}^{k} p_j, \quad k = 1, \ldots, N. \quad (7)$$

Because $p_j > 0$, this construction guarantees coefficients $\boldsymbol{\beta}$ a non-decreasing. As a result, the resulting Bernstein polynomial is guaranteed to be monotonic and properly normalized on $[0,1]$ without any auxiliary regularization.

**Wider or Deeper: The Case for Cascaded Designs.** Although a single MBP is theoretically capable of approximating any function in $\mathcal{H}_{\mathrm{mono}}$ when $N \to \infty$, increasing polynomial degree alone is impractical. Since Bernstein converges slowly, meaning that very high degrees are required to capture sharp radiometric transitions (Lorentz, 2012). In addition, high-degree polynomials rely on high-dimensional Softmax parameterizations, which suffer from vanishing

gradients and inefficient optimization.

To improve expressivity while retaining stable and efficient optimization, we adopt a compositional design that cascades multiple low-degree MBPs instead of increasing polynomial degree. The resulting **cascaded monotonic Bernstein** (CaMB) operator composes several lightweight MBPs:

$$\mathcal{M}_{\text{CaMB}}(z) \triangleq \mathcal{B}_N^{(K)} \circ \mathcal{B}_N^{(K-1)} \circ \cdots \circ \mathcal{B}_N^{(1)}(z). \quad (8)$$

This cascaded structure significantly enhances representational capacity through depth, while preserving monotonicity and favorable optimization behavior. We further show that CaMB can approximate any continuous monotonic function using a compact parameterization (see *proof* in Appendix B.2), making it a principled and effective choice for approximating the unknown forward operator in UDRC.

### 4.2. Joint Optimizing Parametric Model and Signal

To solve the zero-shot UDRC problem, we adopt a plug-and-play (PnP) diffusion framework to jointly optimize desired signals and the parametric forward model. We formulate the recovery as the following joint optimization problem:

$$\hat{x}, \hat{\Theta} = \arg\min_{x, \Theta} \|y - \mathcal{M}_{\Theta}(x)\|_2^2 - \lambda \cdot \log p(x), \quad (9)$$

where $\mathcal{M}_{\Theta}$ is the CaMB-parameterized UDRC forward model, $\Theta$ represents its parameters, and $p(x)$ is the data prior provided by pre-trained diffusion models.

**HQS Decomposition.** Within PnP-Diffusion framework, we employ the half-quadratic splitting (HQS) scheme, introducing auxiliary variable $z$ to decouple the data fidelity term from the prior term. The resulting optimization is solved by alternating between the following two subproblems:

$$\begin{cases} z^i, \Theta^i = \arg\min_{z, \Theta} \|y - \mathcal{M}_{\Theta}(z)\|_2^2 + \dfrac{1}{\eta^2}\|z - x^{i-1}\|_2^2, \\ x^i = \arg\min_{x} \dfrac{1}{2\eta^2}\|x - z^i\|_2^2 - \lambda \log p(x). \end{cases}$$
$$(10)$$

where $\eta$ controls the coupling strength between $x$ and $z$.

**Solving the Subproblems.** The data-fidelity subproblem is solved via gradient-based optimization, since the CaMB operator $\mathcal{M}_{\Theta}$ is fully differentiable with respect to both $z$ and $\Theta$. The prior subproblem can be solved by directly applying the pre-trained diffusion model as a denoiser:

$$x^i = \mathcal{D}_{\theta}(x_t, t), \quad x_t = z^i + \sqrt{1 - \bar{\alpha}_t}\,\epsilon, \quad (11)$$

where $\mathcal{D}_{\theta}(\cdot, t)$ denotes the diffusion denoiser at timestep $t$, $\bar{\alpha}_t$ is the cumulative noise schedule, and $\epsilon \sim \mathcal{N}(0, I)$. The optimization process is illustrated in Fig. 2 and Algorithm 1. After optimization, high-quality images are recovered under

---

**Algorithm 1** CaMB-Diff

**input** Pre-trained diffusion model $\mathcal{D}_{\theta}$, total iterations $I$, paramtric forward model $\mathcal{M}_{\Theta}$, measurement $y$, noise schedule $\{\sigma_t\}_{t=1}^T$, guidance scale $\lambda_t$, step size $\eta$.
1: **Initialize:** $x^0, z^0 \sim \mathcal{N}(0, I)$, $\Theta = I$.
2: **for** $i = 0$ to $I$ **do**
3:     // Set diffusion noise schedule
4:     $t \leftarrow t_i$ (*e.g.*, linear spacing from $T$ to 1)
5:     ▷ *1. Data Fidelity Optimization*
6:     **for** $j = 1$ **to** $J$ **do**
7:         $\mathcal{L} = \|y - \mathcal{M}_{\Theta}(z)\|^2 + \lambda_t\|z - x^i\|^2,$
8:         $z = z - \eta\nabla_z\mathcal{L}, \quad \Theta = \Theta - \eta\nabla_{\Theta}\mathcal{L}$
9:     **end for**
10:    ▷ *2. Prior Optimization via Denoising*
11:    $x_{t_i} = z_i + \sqrt{1 - \bar{\alpha}_{t_i}}\,\epsilon, \epsilon \sim \mathcal{N}(0, I),$
12:    $x^i = \mathcal{D}_{\theta}(x_{t_i}, t_i),$
13: **end for**
14: **return** $z^I$

---

CaMB's physical constraints, with diffusion providing a strong geometric prior for structural and semantic recovery.

**Diffusion Guidance Schedule.** In the standard HQS framework (Zhu et al., 2023), the schedule is derived as $\lambda_t = \lambda\bar{\alpha}_t/(1 - \bar{\alpha}_t)$. We observe that this formulation assigns excessive weight to the data consistency term during the early stages of reverse sampling, causing both the operator and the latent image to prematurely converge to a trivial solution. To resolve this, we propose a U-shaped scheduling strategy:

$$\lambda_t = \lambda\frac{\bar{\alpha}_t}{1 - \bar{\alpha}_t} + C\left(\frac{t}{T}\right)^k. \quad (12)$$

This schedule facilitates a dynamic three-stage optimization: (1) In early stages, the prior dominates to anchor global illumination and prevent early overfitting; (2) In middle stages,, the data consistency term takes precedence to reconstruct structural details; (3) In late stages, the prior dominates again to refine the structures on the natural image manifold.

## 5. Experiments & Results

### 5.1. Experimental Setup

To comprehensively evaluate CaMB-Diff, we consider three tasks spanning two regimes of signal intensity degradation: **signal starvation** (low-light image and low-field MRI enhancement) and **signal saturation** (HDR reconstruction).

**Datasets.** In **low-light enhancement** task, we employed the LOLv1 (Wei et al., 2018) and LOLv2 (Yang et al., 2021) benchmarks. The LOLv2 dataset is explicitly composed of real and synthetic subsets. In **low-field MRI enhancement** task, we utilized the 3T high-field HCP dataset (Van Es-

*Table 1.* **Quantitative comparison of low-light image enhancement on the LOLv1, LOLv2-real, and LOLv2-synthetic benchmarks.** The method types "S" and "Z" denote Supervised and Zero-shot methods, respectively. For clarity, the best overall results are underlined, while the best zero-shot performance is highlighted in **bold**.

| Type | Methods | LOL v1 (Wei et al., 2018) | | | | | LOL v2 Real (Yang et al., 2021) | | | | | LOL v2 Synthetic (Yang et al., 2021) | | | | |
|---|---|---|---|---|---|---|---|---|---|---|---|---|---|---|---|---|
| | | PSNR↑ | SSIM↑ | LPIPS↓ | FID↓ | LOE↓ | PSNR↑ | SSIM↑ | LPIPS↓ | FID↓ | LOE↓ | PSNR↑ | SSIM↑ | LPIPS↓ | FID↓ | LOE↓ |
| S | RetinexNet (Wei et al., 2018) | 18.21 | 0.7962 | 0.3195 | 130.13 | 445.75 | 18.12 | 0.8052 | 0.3438 | 142.89 | 595.90 | 17.59 | 0.8123 | 0.2399 | 86.30 | 534.90 |
| | KinD (Zhang et al., 2019) | 20.38 | 0.9076 | 0.1586 | 63.25 | 437.55 | 18.63 | 0.8821 | 0.1455 | 63.08 | 445.97 | 17.65 | 0.8695 | 0.2599 | 85.56 | 436.02 |
| | Restormer (Zamir et al., 2022) | 22.92 | 0.9314 | 0.1655 | 76.44 | 134.35 | 19.04 | 0.8891 | 0.2013 | 72.50 | 233.57 | 21.92 | 0.9183 | 0.1579 | 45.51 | 183.17 |
| | SNR-Net (Xu et al., 2022) | 25.71 | 0.9426 | 0.1422 | 54.83 | 188.15 | 22.11 | 0.9251 | 0.1434 | 52.06 | 206.59 | 24.48 | 0.9528 | 0.0651 | 18.61 | 204.31 |
| | Retinexformer (Cai et al., 2023) | 26.20 | 0.9498 | 0.1554 | 65.46 | 150.06 | 23.32 | 0.9234 | 0.1656 | 58.00 | 199.79 | 26.18 | 0.9607 | 0.0744 | 20.71 | 186.08 |
| | HVI (Yan et al., 2025) | 24.19 | 0.9395 | 0.1143 | 43.17 | 145.69 | 24.72 | 0.9406 | 0.1312 | 50.52 | 153.12 | 26.17 | 0.9684 | 0.0544 | 16.35 | 161.37 |
| Z | Zero-DCE (Guo et al., 2020) | 16.91 | 0.7600 | 0.2712 | 101.11 | 245.54 | 18.06 | 0.7027 | 0.2795 | 80.37 | 201.99 | 17.36 | 0.8128 | 0.2782 | 89.39 | 218.86 |
| | RUAS (Liu et al., 2021) | 16.79 | 0.7771 | 0.2612 | 103.87 | 153.18 | 15.67 | 0.7504 | 0.2849 | 88.05 | 174.62 | 16.65 | 0.7201 | 0.3742 | 112.76 | 491.27 |
| | GDP (Fei et al., 2023) | 17.06 | 0.7145 | 0.3581 | 126.39 | 190.22 | 15.05 | 0.6265 | 0.3965 | 118.95 | 278.67 | 12.13 | 0.4934 | 0.2873 | 94.24 | 75.99 |
| | TAO (Gou et al., 2024) | 17.18 | 0.8099 | 0.4153 | 127.32 | 331.13 | 18.98 | 0.8019 | 0.3482 | 125.88 | 440.91 | 13.30 | 0.6135 | 0.4783 | 120.57 | 288.14 |
| | FourierDiff (Lv et al., 2024) | 17.85 | 0.8340 | 0.2575 | 79.35 | 166.71 | 16.23 | 0.7970 | 0.2648 | 76.31 | 121.82 | 14.13 | 0.6682 | 0.2590 | 77.19 | 130.92 |
| | AGLLDiff (Lin et al., 2025) | 19.57 | **0.8699** | 0.2701 | 123.90 | 225.22 | 19.46 | **0.8543** | 0.2700 | 120.15 | 265.29 | 18.81 | 0.8735 | 0.2572 | 93.73 | 299.93 |
| | CaMB-Diff (Ours) | **20.14** | 0.8498 | **0.2171** | **72.65** | **82.21** | **19.58** | 0.8359 | **0.2177** | **66.90** | **88.27** | **19.08** | **0.8748** | **0.1692** | **48.95** | **38.85** |

*Figure 3.* **Qualitative comparison of low-light image enhancement on the LOLv1, LOLv2-real, and LOLv2-synthetic benchmarks.**

sen et al., 2013), consisting of skull-stripped volumes at 1mm isotropic resolution. The dataset was partitioned into 70%/15%/15% for training, validation, and testing, where the training split was used exclusively to learn the clean high-field diffusion prior. Realistic low-field observations were synthesized from the high-field test split following the physics-driven model in (Lin et al., 2023), simulating field-dependent SNR decay and contrast shifts calibrated to match real-world intensity statistics. In **HDR Reconstruction**, we evaluated on 100 images of the ImageNet validation set (Deng et al., 2009). To simulate sensor saturation, we applied a hard clipping operation on the normalized inputs $\mathbf{x} \in [-1, 1]$ via $\mathbf{y} = \text{clip}(2\mathbf{x}, -1, 1)$ (Zhang et al., 2025). Additionally, Gaussian noise with $\sigma = 0.05$ was added to model measurement uncertainty. To further evaluate robustness in real-world scenarios, we extend our evaluation to the HDR-Eye dataset (Nemoto et al., 2015), which contains real captured images with complex multi-exposure ISP effects.

**Baselines.** We compare CaMB-Diff against two categories

of methods: (1) **Universal Zero-Shot Solvers**, benchmarking against GDP (Fei et al., 2023) and TAO (Gou et al., 2024) across all tasks to isolate the impact of operator modeling; and (2) **Task-Specific Specialists**, covering SOTA in each domain. For low-light image enhancement, we compare with supervised methods: RetinexNet (Wei et al., 2018), KinD(Zhang et al., 2019), Restormer (Zamir et al., 2022), SNR-Net (Xu et al., 2022), Retinexformer (Cai et al., 2023), and the latest supervised SOTA, HVI (Yan et al., 2025) and unsupervised/zero-shot methods: Zero-DCE (Guo et al., 2020), RUAS (Liu et al., 2021), FourierDiff (Lv et al., 2024), AGLLDiff (Lin et al., 2025)). For low-field MRI enhancement, we evaluate against supervised baseline PF-SR (Man et al., 2023), unsupervised baseline ULFInr (Islam et al., 2025), and zero-shot baseline DiffDeuR (Lin et al., 2024).

**Metrics.** Reconstruction quality is assessed via PSNR and SSIM for signal fidelity, and LPIPS (Zhang et al., 2018)/FID (Heusel et al., 2017) for perceptual realism. For low-light tasks, we additionally report the Lightness Order

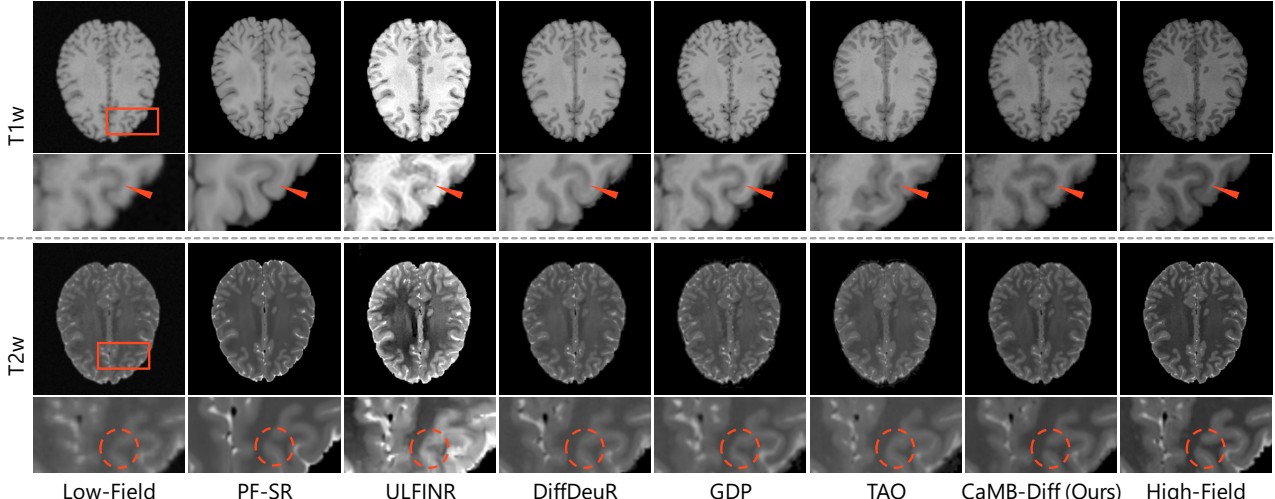

*Figure 4.* **Qualitative comparison of low-field MRI enhancement on the HCP dataset.**

*Table 2.* **Quantitative comparison of low-field MRI enhancement on the HCP dataset.** The best results are highlighted in **bold**.

| Method | T1w | | | T2w | | |
|---|---|---|---|---|---|---|
| | PSNR↑ | SSIM↑ | LPIPS↓ | PSNR↑ | SSIM↑ | LPIPS↓ |
| ULFInr (Islam et al., 2025) | 22.06 | 0.7172 | 0.2194 | 20.26 | 0.5689 | 0.2699 |
| PF-SR (Man et al., 2023) | 21.92 | 0.7052 | 0.2126 | 21.21 | 0.6542 | 0.2001 |
| DiffDeuR (Lin et al., 2024) | 23.17 | 0.6840 | 0.2611 | 23.66 | 0.8087 | 0.2620 |
| GDP (Fei et al., 2023) | 23.42 | 0.6789 | 0.2169 | 23.92 | 0.7657 | 0.2278 |
| TAO (Gou et al., 2024) | 23.61 | 0.6892 | 0.2156 | 24.18 | 0.7821 | 0.2094 |
| CaMB-Diff (Ours) | **26.43** | **0.8020** | **0.1602** | **25.39** | **0.8141** | **0.1404** |

Error (LOE) (Wang et al., 2013) to quantify the preservation of natural illumination order.

**Implementation Details.** We utilized unconditional ADM (Dhariwal & Nichol, 2021) pre-trained on ImageNet for natural image tasks (Low-Light/HDR) and trained a domain-specific DDPM on the HCP training split for MRI. Inference is accelerated via DDIM (Song et al., 2020a) with 100 sampling steps for all tasks. In each reverse timestep, the joint optimization is solved using Adam with 20 internal iterations (*i.e.*, $J = 20$) and $\eta = 0.01$.

### 5.2. Main Results

**Low-Light Enhancement.** Quantitative comparisons in Table 1 demonstrate that CaMB-Diff outperforms all zero-shot baselines in perceptual metrics(LPIPS/FID) while maintaining signal fidelity (PSNR/SSIM) competitive with sota zero-shot task-specific methods. Crucially, our method achieves the lowest Lightness Order Error (LOE) across all categories, empirically validating the necessity of monotonicity for preserving natural illumination. Qualitative comparison is shown in Fig. 3. While the high-bias parametric modeling of GDP introduces blocking artifacts and the unconstrained optimization of TAO leads to severe color shifts,

CaMB-Diff distinctly surpasses these approaches, delivering aesthetically natural enhancements with optimal brightness, accurate color fidelity, and restored global contrast.

**Low-Field MRI Enhancement.** Quantitative assessments in Table 2 establish CaMB-Diff's superiority, outperforming all baselines across both fidelity (PSNR/SSIM) and perceptual (LPIPS/FID) metrics. Visual comparison in Fig. 4 further highlights this advantage: our method not only restores the high-contrast dynamics characteristic of high-field imaging but also faithfully resolves anatomical details obscured in the low-field input.

**HDR Reconstruction.** Quantitative comparisons in Table 3 demonstrate that CaMB-Diff significantly outperforms all zero-shot baselines. Our method achieves substantial margins across both signal fidelity (PSNR/SSIM) and perceptual metrics (LPIPS/FID), surpassing the nearest competitor by over 3dB in PSNR. Visual comparisons in Fig. 5 further illustrate that our method not only restores the global high dynamic range but also successfully hallucinates plausible high-frequency details in hard-clipped regions, effectively resolving the saturation ambiguity where other methods fail. To evaluate the robustness of our method against unknown ISP pipelines, we validate our approach on the real-world HDR-Eye dataset. As reported in Table 3, CaMB-Diff consistently outperforms zero-shot baselines. Qualitative results demonstrate that our approach successfully restores hidden details in extreme shadows while refining textures in brighter regions without overexposure, further confirming its practicality in authentic imaging scenarios.

**Out-of-Distribution Robustness.** Out-of-Distribution (OOD) robustness is a core advantage of zero-shot methods. While supervised approaches achieve high in-domain performance, they are fundamentally constrained by the dis-

*Table 3.* Quantitative comparison of HDR Reconstruction. The evaluation spans both synthetic (ImageNet) and real-world (HDR-Eye) datasets. The best results are highlighted in **bold**.

| Dataset | Method | PSNR↑ | SSIM↑ | LPIPS↓ |
|---|---|---|---|---|
| ImageNet (Synthetic) | GDP (Fei et al., 2023) | 16.80 | 0.7003 | 0.3016 |
| | TAO (Gou et al., 2024) | 16.77 | 0.6692 | 0.3478 |
| | CaMB-Diff (Ours) | **19.95** | **0.7909** | **0.2639** |
| HDR-Eye (Real) | GDP (Fei et al., 2023) | 15.35 | 0.6950 | 0.1910 |
| | TAO (Gou et al., 2024) | 14.82 | 0.6420 | 0.3710 |
| | CaMB-Diff (Ours) | **17.78** | **0.7880** | **0.1880** |

*Table 4.* **Quantitative comparison of OOD performance.** Models are trained on LOLv2-synthetic and tested on LOLv1.

| Method | PSNR↑ | SSIM↑ | LPIPS↓ | FID↓ | LOE↓ |
|---|---|---|---|---|---|
| Retinexformer (Cai et al., 2023) | 11.54 | 0.604 | 0.283 | 95.53 | 199.95 |
| HVI (Yan et al., 2025) | 13.63 | 0.727 | 0.315 | 108.08 | 185.14 |
| CaMB-Diff (Ours) | **20.14** | **0.849** | **0.217** | **72.65** | **82.21** |

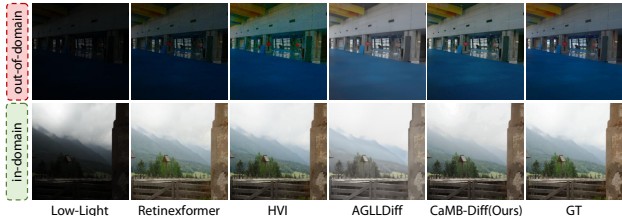

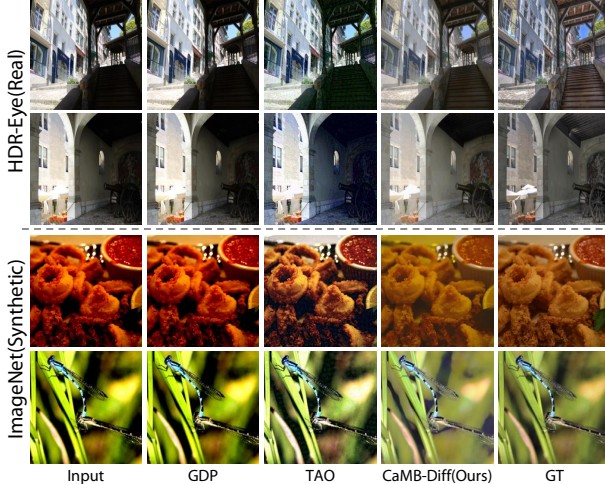

*Figure 5.* **Qualitative comparison of HDR reconstruction on the ImageNet dataset and HDR-Eye dataset.**

*Figure 6.* **Qualitative results of three baselines and CaMB-Diff for low-light enhancement on out-of-domain test.** The supervised baselines(Retinexformer, HVI) are trained on LOLv2 synthetic dataset and the out-of-domain images are sampled from LOLv1 dataset.

applied to the ground truth and the actual observation.

As observed in the left part of Fig. 8, baseline methods including DiffDeuR, GDP, and TAO successfully minimize the fidelity error, rapidly converging to a low value similar to ours. However, a critical disparity emerges in the right part of Fig. 8 regarding the validation error. The unconstrained methods (GDP, TAO) fail to converge to a low error state, exhibiting a significant gap between training fidelity and ground-truth validation. This reveals a fundamental optimization-generalization gap: without rigorous topological constraints, these models exploit their excess capacity to overfit specific noise realizations or learn physically implausible shortcuts to satisfy data consistency. In contrast, our method achieves stable convergence to a minimal error state via a monotonic descent. By imposing strict monotonicity as a hard inductive bias, our framework confines the search space to the manifold of valid radiometric mappings. This ensures that the minimization of data fidelity acts as a reliable proxy for recovering the true physical operator, effectively preventing the overfitting pathology common in blind inverse solvers.

**Effectiveness of Cascaded Design.** While the Bernstein approximation theorem posits that a single polynomial of sufficiently high degree $N$ can approximate any continuous function, we argue that a *cascaded composition* of low-degree polynomials is superior for gradient-based optimization. We validate this architectural choice quantitatively and qualitatively in Fig. 7. Results indicate that increasing the width (degree) of a single layer is inefficient. Specifically, the configuration with a single high-degree layer ($N = 64, K = 1$) yields suboptimal performance (10.99 dB). As visualized in Fig. 7 (2nd column), this variant fails to lift the signal from the noise floor, resulting in an unen-

tribution of their training data. To validate this, we conduct an OOD evaluation: models trained on the LOLv2-synthetic dataset are directly applied to the real-world LOLv1 benchmark. As shown in Table 4, supervised methods suffer from significant performance degradation due to domain shifts. In contrast, CaMB-Diff dynamically optimizes the forward operator on the fly without requiring degradation-specific paired training data, successfully outperforming the supervised state-of-the-art method HVI in the OOD evaluation. This highlights the strong generalization capability of our structured operator modeling in practical, unpredictable lighting conditions.

### 5.3. Ablation Studies

**Effectiveness of Structured Forward Operator Modeling.** To strictly validate whether our method learns the authentic physical degradation rather than overfitting the measurement, we track the convergence dynamics of the forward operator estimation on the Low-Field MRI task. We monitor two complementary metrics in Fig. 8: (1) **fidelity error** ($\|\boldsymbol{y} - \mathcal{M}_{\Theta}(\boldsymbol{z})\|^2$), which guides the zero-shot optimization, and (2) **validation Error** ($\|\boldsymbol{y} - \mathcal{M}_{\Theta}(\boldsymbol{x}_{\text{GT}})\|^2$), an oracle metric evaluating the error between the learned operator

*Table 5.* **Ablation study on architectural design and parameter efficiency.** We evaluate the impact of operator depth ($K$) versus width (degree $N$) on the LOLv1 dataset. Our cascaded design ($N = 3, K = 8$) achieves superior fidelity with minimal parameters, distinctly outperforming the unstable high-degree variant ($N = 64$) and heavy baselines.

| Forward Model | Num of Params | LOL v1 | | | | |
|---|---|---|---|---|---|---|
| | | PSNR | SSIM | LPIPS | FID | LOE |
| CaMB(N=64 K=1) | 63 | 10.99 | 0.5165 | 0.2627 | 95.60 | **49.63** |
| CaMB(N=2 K=8) | **8** | 19.94 | 0.8476 | 0.2311 | 87.91 | 82.41 |
| CaMB(N=3 K=8) | 16 | **20.14** | **0.8498** | **0.2171** | **72.65** | 82.21 |
| GDP(fx+M) | 65537 | 17.06 | 0.7145 | 0.3581 | 126.39 | 190.22 |
| TAO(CNNs) | 742660 | 17.18 | 0.8099 | 0.4153 | 127.32 | 331.13 |

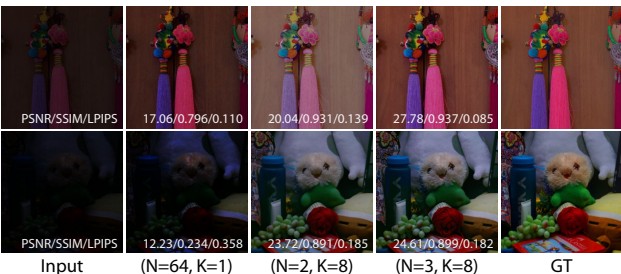

Input    (N=64, K=1)    (N=2, K=8)    (N=3, K=8)    GT

*Figure 7.* **Qualitative and Quantitative comparisons of CaMB-Diff with various cascaded designs.** While the single high-degree model ($N = 64, K = 1$) fails to enhance the dark input, our cascaded low-degree design ($N = 3, K = 8$) effectively recovers the illumination and details.

hanced dark output. Theoretically, optimizing high-degree Bernstein polynomials involves determining coefficients via a high-dimensional Softmax ($\mathbb{R}^{64}$), which exacerbates gradient vanishing issues and results in a rugged optimization landscape that hinders convergence. Conversely, our default cascaded design ($N = 3, K = 8$) achieves state-of-the-art performance with significantly fewer parameters. The visual comparison in Fig. 7 further corroborates that this setting restores natural illumination and vivid details. This confirms the advantage of the "Deeper-over-Wider" strategy: by composing lightweight atomic operators, CaMB achieves an effective algebraic degree of $N^K$ (exponential expressivity) while maintaining a benign optimization landscape suitable for zero-shot adaptation.

## 6. Conclusion

In this paper, we presented CaMB-Diff, a unified zero-shot framework for resolving Inverse Problems under unknown dynamic range compression (DRC). By identifying the common mathematical structure behind diverse physical degradations—from signal starvation in low-field MRI to saturation in HDR imaging—we proposed the Cascaded Monotonic Bernstein (CaMB) operator. This novel parameterization structurally embeds the physical law of monotonicity into the forward model, effectively resolving the dilemma between expressivity and optimization stability inherent

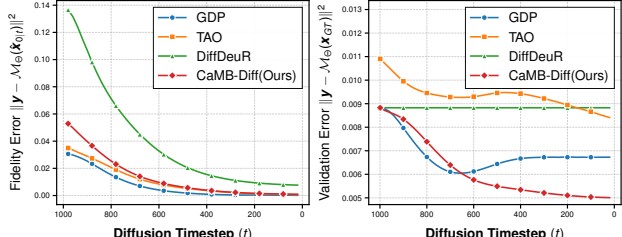

*Figure 8.* **Convergence Analysis of Operator Estimation.** We track the dynamics of the learned forward operator across the reverse diffusion process. **(Left) Fidelity Error:** Most methods successfully minimize the data consistency objective towards the observed measurement $\boldsymbol{y}$. **(Right) Validation Error:** A critical generalization gap emerges when evaluating against the ground truth. Unconstrained methods (GDP, TAO) fail to converge to a low validation error. In contrast, **CaMB-Diff** achieves stable convergence to a minimal error state via a monotonic descent.

in unknown blind solvers. Integrated with generative diffusion priors, our framework successfully recovers latent signals from regimes of severe information loss without paired supervision. Extensive experiments demonstrate that CaMB-Diff not only achieves state-of-the-art fidelity but also ensures rigorous physical consistency, offering a robust paradigm for reliable radiometric recovery in computational imaging.

## Impact Statement

This paper presents work whose goal is to advance the field of Machine Learning. There are many potential societal consequences of our work, none which we feel must be specifically highlighted here.

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

# A. Experimental Details

## A.1. Baseline Details

### A.1.1. DIFFUSION BASED METHODS

**GDP** All experiments are conducted with the original code and default settings as specified in (Fei et al., 2023). For low-field MRI enhancement, we set the guidance scale = 80000.

**TAO** All experiments are conducted with the original code and default settings as specified in (Gou et al., 2024). For low-field MRI enhancement and HDR reconstruction, we set the guidance scale = 50000.

**AGLLDiff** Following (Lin et al., 2025), all experiments are conducted using the original code and default settings. Images are cropped to $256 \times 256$ to be compatible with the pre-trained diffusion prior.

**FourierDiff** Following (Lv et al., 2024), all experiments are conducted using the original code and default settings. Images are cropped to $256 \times 256$ to be compatible with the pre-trained diffusion prior.

**DiffDeuR** We adopt the default setting in (Lin et al., 2024) with 100 DDIM steps.

### A.1.2. LEARNING BASED METHODS

For low-light enhancement comparisons (**Zero-DCE** (Guo et al., 2020), **RUAS** (Liu et al., 2021), **RetinexNet** (Wei et al., 2018), **KinD** (Zhang et al., 2019), **Restormer** (Zamir et al., 2022), **SNR-Net** (Xu et al., 2022), **Retinexformer** (Cai et al., 2023), **HVI** (Yan et al., 2025)), we employ the official codes with default configurations, training them on the LOLv1 and LOLv2 datasets. For low-field MRI, we train the supervised baseline **PF-SR** (Man et al., 2023) on the HCP dataset using the proposed degradation simulation and its default hyperparameters. For **ULFInr** (Islam et al., 2025), we adhere to its default settings, employing registered 3T high-field images as reference.

## A.2. Low-Field MRI Simulation Details

Since paired high-field (HF) and low-field (LF) MRI data from the same subject are scarce and often misaligned, we synthesize realistic LF observations from HF scans following the physics-driven stochastic simulator proposed in (Lin et al., 2023). This simulation pipeline explicitly models the two fundamental degradations inherent to low-field acquisition: *anisotropic geometric loss* and *radiometric contrast distortion*.

**High-Field Source Data.** We utilize the 3T high-field T1-weighted (T1w) images from the Human Connectome Project (HCP) dataset (Van Essen et al., 2013), acquired with 0.7mm isotropic resolution. Skull stripping is performed to focus on brain tissue.

**Physics-driven Degradation Process.** The simulation $\mathbf{y} = \mathcal{M}(\mathbf{x}) + \mathbf{n}$ proceeds in three stages:

1. **Geometric Downsampling (Thick Slices):** To simulate the low-through-plane resolution typical of LF scanners, we apply a 1D Gaussian blurring kernel along the slice direction ($z$-axis) followed by downsampling. We set the target slice thickness to 5mm with a 1mm gap, mimicking the clinical protocol of the 0.36T MagSense 360 scanner (Lin et al., 2023).

2. **Radiometric Contrast Transfer:** Unlike natural images where intensity scales linearly, MRI contrast depends on field-strength-dependent tissue relaxation times ($T_1, T_2$). We employ a tissue-specific intensity mapping. First, tissue segmentation masks (White Matter, Grey Matter, CSF) are generated from the HF source using SPM. Then, for each tissue class $c$, the HF intensity is scaled by a factor $\nu_c = \mu_c^{LF}/\mu_c^{HF}$, where $\mu_c^{LF}$ is derived from the theoretical signal equation at 0.36T. This step introduces the complex, non-linear (yet monotonic) dynamic range compression that our CaMB operator aims to correct.

3. **Noise Injection:** Finally, we inject Gaussian-distributed noise to match the low SNR profile of LF scanners. The noise levels are calibrated such that the synthesized images match the SNR distribution of real clinical scans from the LF17 dataset (Lin et al., 2023).

This rigorous simulation ensures that the synthetic data preserves the topological structure of the brain while exhibiting the authentic *signal starvation* and *contrast compression* artifacts observed in real-world low-field MRI.

### A.3. Metrics Details

**Standard Metrics.** We assess reconstruction quality using Peak Signal-to-Noise Ratio (PSNR) and Structural Similarity Index (SSIM) for signal fidelity, and Learned Perceptual Image Patch Similarity (LPIPS) (Zhang et al., 2018) along with Fréchet Inception Distance (FID) (Heusel et al., 2017) for perceptual realism. All standard metrics are computed using the official implementations in the `torchmetrics` library.

**Lightness Order Error (LOE).** Crucially, to validate the physical consistency of the learned radiometric mapping, we employ the Lightness Order Error (LOE) (Wang et al., 2013). Unlike pixel-wise error metrics, LOE quantifies the preservation of the *relative intensity order* between the enhanced image and the reference, serving as a direct empirical validation for our monotonicity constraint. The LOE is defined as:

$$\text{LOE} = \frac{1}{M} \sum_{i=1}^{M} \sum_{j=1}^{M} \left( U(L_i, L_j) \oplus U(L_i^{\text{ref}}, L_j^{\text{ref}}) \right), \tag{13}$$

where $M$ is the total number of pixels, and $\oplus$ denotes the exclusive-OR (XOR) operator. The indicator function $U(p, q)$ returns 1 if $p \geq q$, and 0 otherwise. $L_k$ and $L_k^{\text{ref}}$ represent the lightness of the $k$-th pixel in the enhanced and reference images, respectively, calculated as the maximum value among R, G, and B channels ($L_k = \max_{(c \in \{R,G,B\})} I_{k,c}$).

Due to the quadratic computational complexity $O(M^2)$ of the pixel-wise comparison, we perform a $4\times$ bicubic downsampling on the images prior to LOE calculation. A lower LOE score indicates that the method better preserves the natural illumination hierarchy and respects the topological ordinality of the signal.

## B. Theoretical Analysis

In this section, we provide rigorous proofs for the properties of the proposed Cascaded Monotonic Bernstein (CaMB) operator. We first establish the fundamental properties of the atomic Monotonic Bernstein Polynomial (MBP) layer—specifically its guaranteed monotonicity (Safety) and approximation capability (Completeness). Subsequently, we prove the universality of the cascaded composition.

### B.1. Properties of the Atomic MBP Layer

Recall that a single MBP layer of degree $N$ is defined as $B_N(z; \boldsymbol{\beta}) = \sum_{k=0}^{N} \beta_k b_{k,N}(z)$, where $b_{k,N}(z) = \binom{N}{k} z^k (1-z)^{N-k}$ are the Bernstein basis polynomials. To enforce structural constraints, the coefficients $\boldsymbol{\beta}$ are parameterized via learnable weights $\mathbf{w} \in \mathbb{R}^N$ and a Softmax operation:

$$\beta_0 = 0, \quad \beta_k = \sum_{j=1}^{k} [\text{Softmax}(\mathbf{w})]_j, \quad \forall k \in [1, N]. \tag{14}$$

**Lemma B.1** (Monotonicity / Safety). *The polynomial $B_N(z; \boldsymbol{\beta})$ parameterized by Eq. 14 is monotonically increasing on the interval $[0, 1]$.*

*Proof.* The derivative of a Bernstein polynomial of degree $N$ is given by a Bernstein polynomial of degree $N - 1$:

$$\frac{d}{dz} B_N(z; \boldsymbol{\beta}) = N \sum_{k=0}^{N-1} (\beta_{k+1} - \beta_k) b_{k,N-1}(z). \tag{15}$$

From our parameterization in Eq. 14, the difference term is:

$$\beta_{k+1} - \beta_k = [\text{Softmax}(\mathbf{w})]_{k+1} = \frac{e^{w_{k+1}}}{\sum_{j=1}^{N} e^{w_j}}. \tag{16}$$

Since the exponential function is strictly positive, we have $\beta_{k+1} - \beta_k > 0$ for all $k$. Furthermore, the Bernstein basis polynomials $b_{k,N-1}(z)$ form a partition of unity and are strictly positive for any $z \in (0, 1)$. Consequently, $\frac{d}{dz} B_N(z; \boldsymbol{\beta})$ is a convex combination of strictly positive values, implying:

$$\frac{d}{dz} B_N(z; \boldsymbol{\beta}) > 0, \quad \forall z \in (0, 1). \tag{17}$$

This guarantees that the learned mapping is monotonic, preventing intensity inversion. □

**Lemma B.2** (Dense Approximation / Completeness). *The set of monotonic Bernstein polynomials is dense in the space of continuous monotonic functions $\mathcal{H}_{mono}$.*

*Proof.* This follows directly from the classical Bernstein approximation theorem. For any continuous function $f \in C[0,1]$, the sequence of polynomials defined by coefficients $\beta_k = f(k/N)$ converges uniformly to $f$ as $N \to \infty$. Crucially, if the target function $f$ is strictly monotonic, the sampled coefficients satisfy $\beta_k < \beta_{k+1}$, ensuring the approximator remains within the monotonic subclass. Thus, the structural constraint does not reduce the theoretical approximation capacity limit as $N \to \infty$. □

## B.2. Proof of Main Theorem: Universality of Cascaded Composition

**Universality of CaMB.** While Lemma B.2 guarantees universality as $N \to \infty$, high-degree polynomials suffer from numerical instability (Runge phenomenon). Here, we prove that a cascade of *low-degree* polynomials can achieve universality, offering a superior parameterization.

**Theorem B.3** (Deep Universality of CaMB). *Let $\mathcal{M}_{CaMB}^{(K,N)}$ denote the hypothesis space of a $K$-layer CaMB operator with fixed degree $N \geq 2$. For any continuous strictly monotonic function $f \in \mathcal{H}_{mono}$ and tolerance $\epsilon > 0$, there exists a depth $K$ such that:*

$$\inf_{\mathcal{M} \in \mathcal{M}_{CaMB}^{(K,N)}} \sup_{z \in [0,1]} |f(z) - \mathcal{M}(z)| < \epsilon. \tag{18}$$

*Proof.* We construct the proof using the concept of **Monotonic Homotopy**. The goal is to decompose a complex global transformation $f$ into a sequence of simple, local deformations close to the Identity mapping.

**1. Construction of the Flow.** Let $I(z) = z$ be the identity map. Since $f(z)$ is strictly monotonic and maps $[0,1]$ to $[0,1]$ (assuming normalized bounds), there exists a continuous homotopy $H : [0,1] \times [0,1] \to [0,1]$ such that $H(z,0) = I(z)$ and $H(z,1) = f(z)$. Specifically, we can use the convex combination $H(z,t) = (1-t)z + tf(z)$, which preserves strict monotonicity for all $t \in [0,1]$.

**2. Layer-wise Discretization.** We partition the "time" interval $[0,1]$ into $K$ steps: $0 = t_0 < t_1 < \cdots < t_K = 1$. The total transformation $f$ can be expressed as the composition of $K$ transition maps $\phi_k$:

$$f = \phi_K \circ \phi_{K-1} \circ \cdots \circ \phi_1, \quad \text{where } \phi_k = H(\cdot, t_k) \circ H^{-1}(\cdot, t_{k-1}). \tag{19}$$

As the number of layers $K \to \infty$, the step size $\Delta t = 1/K$ becomes infinitesimal. The transition map $\phi_k$ approaches the identity:

$$\phi_k(z) \approx z + \delta_k(z), \tag{20}$$

where $\delta_k(z)$ is a microscopic monotonic perturbation.

**3. Local Approximation via Low-Degree Polynomials.** The core insight is that while fitting a complex $f$ requires a high degree $N$, fitting a microscopic perturbation $\phi_k$ only requires low-degree flexibility. We verify that an MBP of degree $N \geq 2$ can approximate Identity and its neighborhood:

- If coefficients are linear ($\beta_j = j/N$), $B_N(z)$ exactly reproduces Identity $z$.

- If $N \geq 2$, the basis contains quadratic terms ($z^2$), allowing non-zero curvature (second derivatives). This enables $B_N$ to approximate the non-linear perturbation $\delta_k(z)$.

Since monotonic polynomials are dense (Lemma B.2), for each sub-function $\phi_k$, there exists an MBP layer $B_N^{(k)}$ that approximates it within error $\epsilon/K$. By the stability of functional composition for Lipschitz continuous functions, the total error of the cascade is bounded.

**Conclusion.** This constructive proof demonstrates that by increasing depth $K$, a sequence of low-degree monotonic polynomials can approximate any strictly monotonic function $f$ with arbitrary precision. This validates the "Depth-for-Width" trade-off: CaMB achieves exponential expressivity ($N^K$ equivalent degree) via linear parameter growth ($K \cdot N$), avoiding the instability of high-degree polynomials. □

## C. Empirical Validation of Assumption 3.1

In Section 3.2, Assumption 3.1 posits that the dynamic range compression $\mathcal{M}$ is spatially stationary and univariate (i.e., the response depends strictly on the signal intensity, independent of spatial position). We conduct an empirical statistical analysis to validate this core assumption.

**Dependence on Signal Intensity:** We extracted paired image data (normal-light target $x$ and degraded observation $y$ ) and mapped all co-located pixel pairs $(x_{i,j}, y_{i,j})$ onto a 2D coordinate system. If Assumption 3.1 holds, these pixel pairs should tightly distribute along a monotonically increasing curve, rather than scattering randomly. As illustrated Fig. 9a, the pixel pair distributions across various datasets densely cluster along a distinct monotonic trajectory, confirming the univariate nature of the degradation.

**Spatial Independence:** To further validate that the degradation is independent of spatial location, we sampled three local patches with entirely distinct textures (*e.g.*, sky, building, and grass) from a single image. Using isotonic regression, we independently fitted three local response curves for the pixel pairs within these respective patches and compared them against the global curve fitted on the entire image. As illustrated Fig. 9b, the four curves exhibited remarkable consistency with negligible variance. This strongly indicates that regardless of variations in local textures or spatial positions, the underlying radiometric degradation experienced by the pixels remains identical, thereby firmly substantiating the spatial stationarity defined in Assumption 3.1.

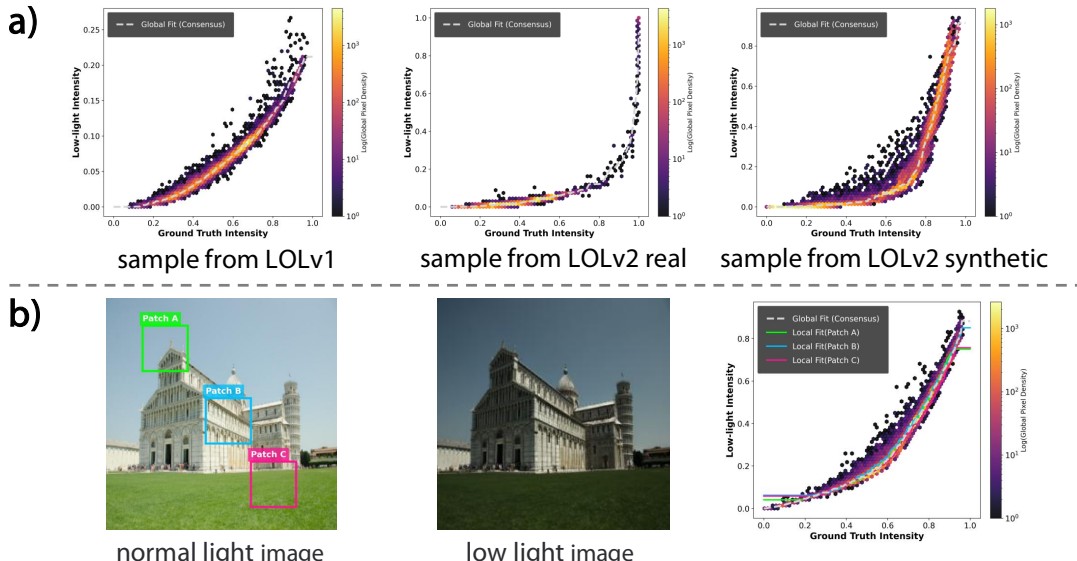

*Figure 9.* **Empirical validation of Assumption 3.1 via joint pixel density analysis.**

## D. Detailed Ablation Studies

### D.1. Why Bernstein Polynomials? Comparison with Monotonic-MLP

To validate the architectural superiority of our Cascaded Monotonic Bernstein (CaMB) operator, we benchmark it against a standard baseline: the Monotonic-MLP. As shown in Table 6, replacing CaMB with a Monotonic-MLP leads to a catastrophic performance drop—degrading PSNR by over 2.2 dB in low-light enhancement and 3.6 dB in HDR reconstruction.

Despite conducting an extensive hyperparameter sweep (varying depth from 2 to 5, width from 8 to 64), we observed two persistent failure modes inherent to the Monotonic-MLP architecture:

1. **Optimization Collapse.** Enforcing strict monotonicity in MLPs typically involves constraining weights to be positive (Wehenkel & Louppe, 2019). We observed that as network depth increases to capture non-linearities, the gradient flow becomes extremely stiff. The positive-weight constraint restricts the optimization path, often causing the network to collapse to a trivial solution (i.e., y=b, where all weights decay towards zero). This creates a "dead" operator that

*Table 6.* **Ablation on Operator Architecture: Monotonic-MLP vs. CaMB.** We compare our proposed cascaded Bernstein formulation against a standard Multi-Layer Perceptron (MLP) with monotonic constraints (positive weights). The "Monotonic-MLP" suffers from optimization stiffness, failing to adapt to the diffusion guidance effectively. In contrast, CaMB provides a smoother, more expressive parameterization, yielding significant gains in both signal fidelity (PSNR) and perceptual quality (FID) across Low-light and HDR tasks.

| Forward Operator | Low-light Image Enhancement | | | | | High Dynamic Range | | | |
|---|---|---|---|---|---|---|---|---|---|
| | PSNR | SSIM | LPIPS | FID | LOE | PSNR | SSIM | LPIPS | FID |
| Monotonic-MLP | 17.94 | 0.7158 | 0.3528 | 237.54 | 132.95 | 16.18 | 0.6245 | 0.3117 | 82.80 |
| CaMB | **20.14** | **0.8498** | **0.2171** | **72.65** | **82.21** | **19.85** | **0.7909** | **0.2639** | **70.73** |

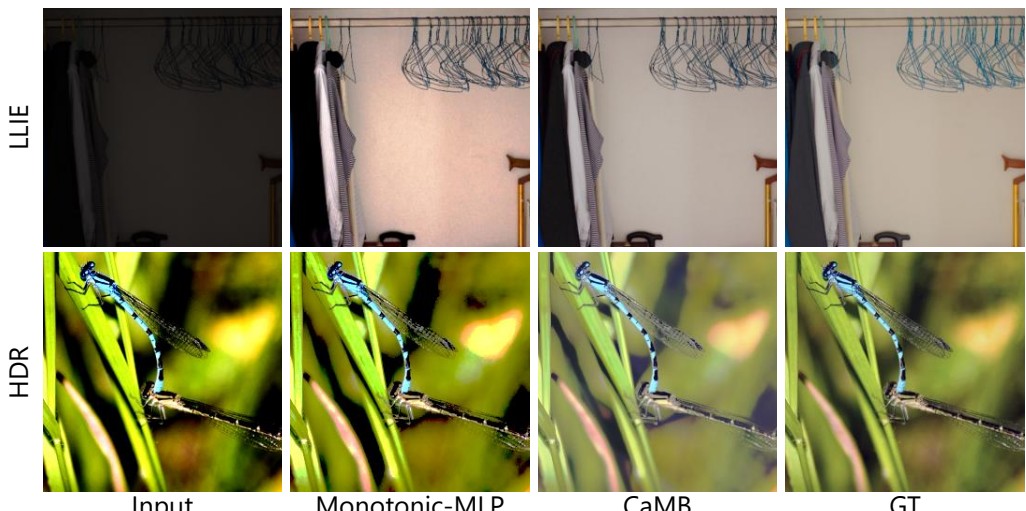

Input          Monotonic-MLP          CaMB          GT

*Figure 10.* **Qualitative ablation of Operator Design on low-light image enhancement(LLIE) and HDR reconstruction(HDR).** While the Monotonic-MLP (2nd column) avoids intensity inversion, its piecewise-linear nature struggles to model the smooth transition into saturation, resulting in flat, lifeless highlights. CaMB (3rd column) leverages the smoothness of Bernstein polynomials to approximate the continuous radiometric response, recovering richer details in the overexposed regions.

fails to respond to the diffusion guidance, resulting in the high bias observed in Table 6.

2. **Lack of Smoothness Inductive Bias.** Radiometric response functions (*e.g.*, camera response curves) are typically smooth (Grossberg & Nayar, 2003) . Bernstein polynomials (the core of CaMB) act as a global smooth approximator, providing a favorable inductive bias for this task. In contrast, Monotonic MLPs (typically using ReLU) produce piecewise linear mappings. Fitting a smooth curvature with piecewise linear segments under blind supervision tends to result in jagged artifacts.

# E. Additional Qualitative Results

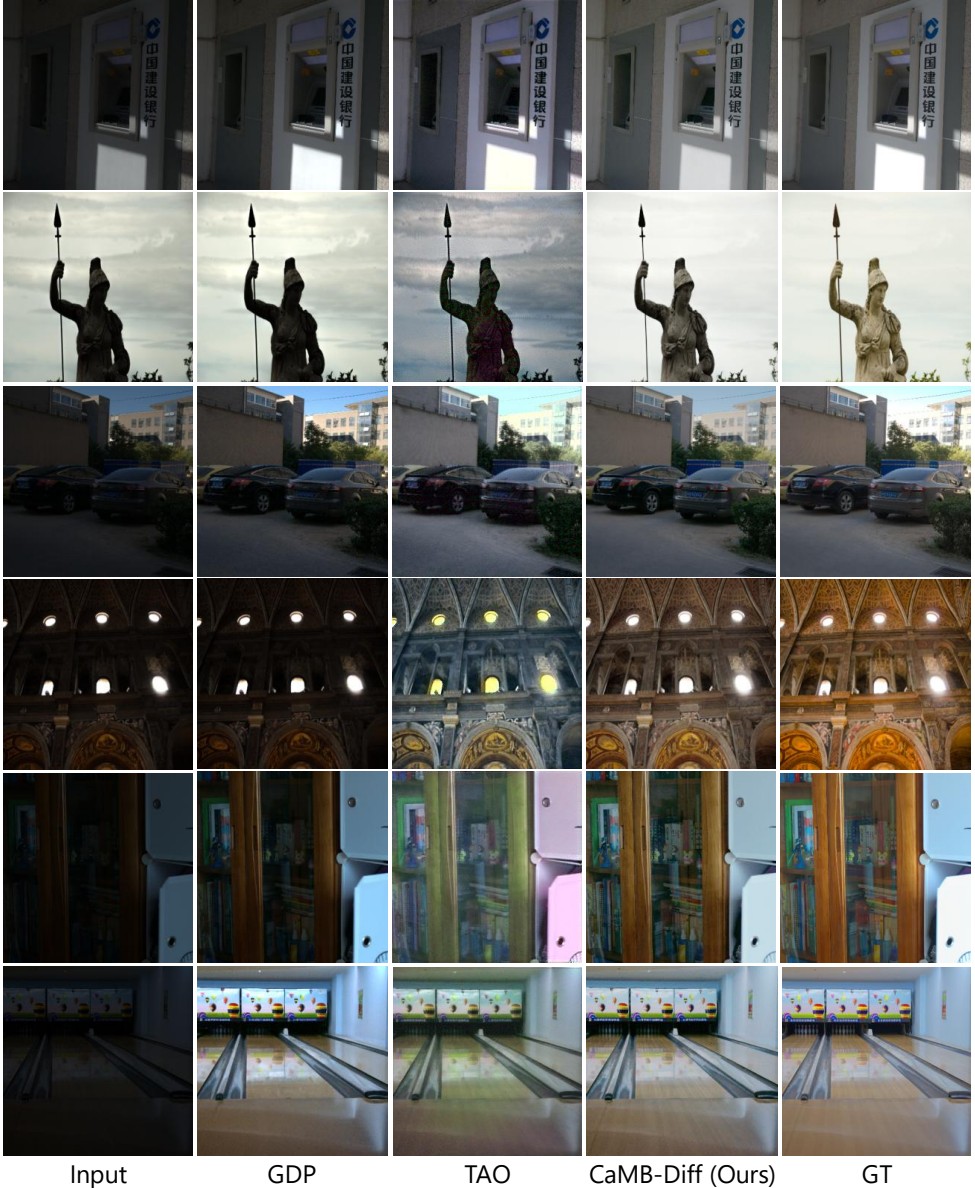

|                |                |                |                   |                |
| Input          | GDP            | TAO            | CaMB-Diff (Ours)  | GT             |

*Figure 11.* **Additional visual comparisons for low-light image enhancement.** We evaluate performance across diverse indoor and outdoor scenes. While GDP often results in under-enhanced contrast and TAO suffers from severe chromatic artifacts (*e.g.*, unnatural color shifts in the statue and cabinet), **CaMB-Diff** consistently restores natural radiance and accurate colors, closely matching the Ground Truth.

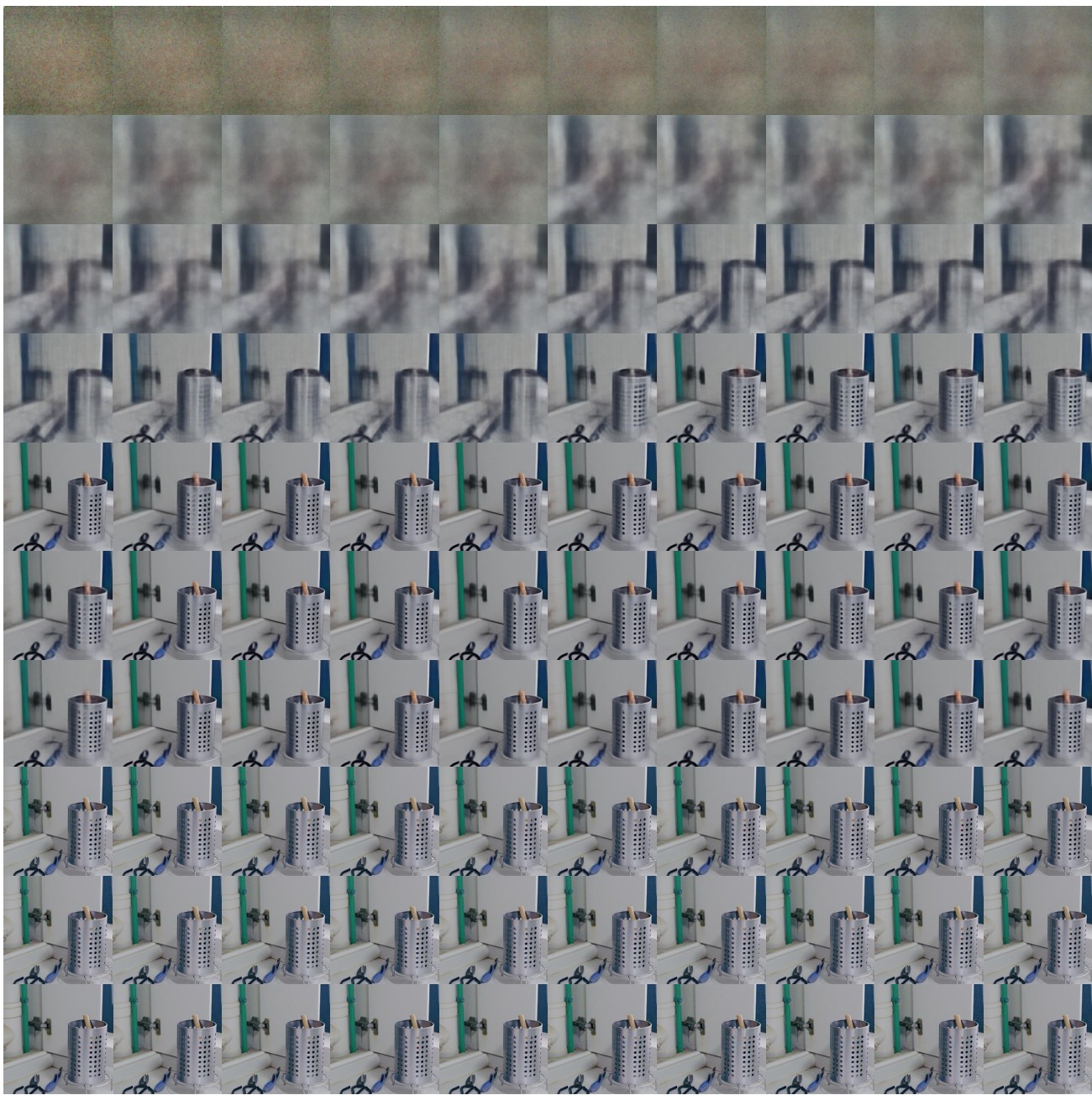

*Figure 12.* **Evolution of the Tweedie Estimate $x_{0|t}$ during Inference.** We visualize the trajectory of the posterior mean estimate $\hat{x}_{0|t}$ across the reverse diffusion process, which demonstrates a stable coarse-to-fine recovery, where the global illumination and object geometry are resolved first, followed by the restoration of fine textures in the saturated regions, guided effectively by our proposed CaMB operator.

