# OpenReview forum: "Resolving Blind Inverse Problems under Dynamic Range Compression via  Structured Forward Operator Modeling"
_ICML.cc/2026/Conference — ICML 2026 regular_

### Official Review · Reviewer_DB5k · 2026-03-09

**Soundness:** 3
**Presentation:** 4
**Significance:** 3
**Originality:** 2
**Overall Recommendation:** 4
**Confidence:** 3

**Summary:**

This paper proposes a method for solving inverse problems arising from unknown dynamic range compression (UDRC), where an observed signal is produced through an unknown nonlinear radiometric mapping of the underlying clean signal. To address this problem, the authors model the unknown forward operator as a point-wise monotonic intensity transformation and introduce the CaMB (Cascaded Monotonic Bernstein) operator to parameterize this mapping. The proposed operator is designed to represent monotonic functions while maintaining stable optimization. The reconstruction framework combines this operator modeling with a plug-and-play diffusion prior, where the unknown signal and the forward operator parameters are jointly optimized at inference time. In the ablation studies, the authors analyze different parameterizations of the Bernstein operator, comparing a single high-degree polynomial with a cascaded design composed of multiple low-degree polynomials. The results show that a deeper cascaded structure provides better optimization behavior and reconstruction quality than a wider single-layer design. Experimental results suggest that the proposed framework performs competitively on several UDRC-related restoration tasks.

**Compliance With Llm Reviewing Policy:**

Affirmed.

**Final Justification:**

The authors’ rebuttal clarified the contributions of the paper more precisely and provided a clearer justification of the proposed approach as an effective solution for inverse problems under Dynamic Range Compression.

**Key Questions For Authors:**

* How broadly does the proposed framework apply beyond the selected UDRC cases? The method is built around the assumption that the unknown forward mapping is monotonic, but the paper does not clearly establish how far this assumption extends beyond low-light, HDR, and low-field MRI.

* When is the monotonicity assumption valid in practice, and under what conditions does it break down?
Since the entire operator design is constrained to the monotonic function class, the paper should better clarify the practical boundary of this assumption and discuss cases where real degradations may not follow a monotonic mapping.

**Limitations:**

While the results are promising on the selected UDRC tasks, the experimental validation is largely confined to a relatively narrow family of monotonic operator settings. As a result, the current evidence does not fully support the paper’s broader framing as a general solution to blind inverse problems.

**Strengths And Weaknesses:**

Strength
* The paper addresses a relevant and practically meaningful problem by formulating low-light enhancement, HDR reconstruction, and low-field MRI recovery under a unified UDRC perspective. In particular, identifying monotonicity as a shared physical invariant across these settings is an interesting and reasonably well-motivated insight.

* The proposed CaMB operator is technically clean: it enforces monotonicity by construction through cascaded Bernstein polynomials, rather than relying on a soft regularizer.
* The method is conceptually well organized, with the diffusion prior handling image recovery and CaMB modeling the unknown radiometric mapping.
* The empirical results are solid within the target setting, and the ablation against Monotonic-MLP helps justify the operator design

Weakness
* The actual scope of the paper seems substantially narrower than the broad framing of “blind inverse problems.” In practice, the method appears tailored to a specific family of problems characterized by unknown but monotonic dynamic-range compression, rather than blind inverse problems more generally. This limits the overall generality and impact of the contribution.
* The core assumption of monotonicity is central to the method, but the paper does not sufficiently discuss when this assumption is valid and when it may break down. As a result, it remains unclear how robust the approach would be beyond the selected UDRC cases.
* While the structured operator design is elegant, the novelty feels somewhat incremental at the system level. The overall framework is essentially a constrained forward-operator parameterization combined with a pre-trained diffusion prior, which may be seen as a careful engineering of known components rather than a fundamentally new inverse-problem paradigm.

---

> ### Author Rebuttal · Authors · 2026-03-31
>
> Thank you for the thoughtful and constructive feedback.
> We appreciate this opportunity to further clarify the contributions of our work and comprehensively discuss its limitations.
> ons. Our point-to-point responses are provided below.
>
> ---
>
> # (Q1 & W1) Scope, Generality, and Impact.
> **Scope**:
> Our goal is NOT to develop a universal framework for all blind inverse problems. Instead, our motivation stems from the observation that tasks such as low-field MRI enhancement, low-light imaging, and HDR reconstruction—despite having disparate underlying physical mechanisms—all adhere to a **shared macroscopic constraint of monotonicity**. Consequently, we define this specific family of tasks as "Blind Inverse Problems under Dynamic Range Compression (uDRC)."
>
> **Generality**:  The root cause of uDRC lies in the fundamental mismatch between signal intensity and the sensor's dynamic range. This pervasive phenomenon extends beyond the image domain (e.g., low-light imaging, HDR reconstruction, and low-field MRI) to encompass 1D signal processing fields (e.g., audio declipping). The ability of our monotonic operator parameterization to unify these cross-disciplinary challenges demonstrates the profound universality of the proposed framework.
>
> **Impact**: In the current field of blind inverse problems, SOTA solvers—whether based on explicit modeling (e.g., BlindDPS[1], GibbsDDRM[2]) or implicit representations (e.g., DIIP[3], DreamClean[4])—predominantly focus on inverse problems caused by **Spatial Degradation** (e.g., blind deblurring and super-resolution). Because these so-called "general" solvers are inherently designed to estimate spatial convolution kernels, they completely fail when applied to uDRC problems.
>
> > [1] Chung, H., & Kim, J.  Parallel diffusion models of operator and image for blind inverse problems, CVPR. 2023
> >
> > [2] Murata, N., et al. GibbsDDRM: A Partially Collapsed Gibbs Sampler for Solving Blind Inverse Problems with Denoising Diffusion Restoration. ICML 2023
> >
> > [3] Chihaoui, H., & Favaro, P.  Diffusion Image Prior, ICCV. 2025
> >
> > [4] Xiao, J., et al. Dreamclean: Restoring clean image using deep diffusion prior. ICLR 2024
>
> # (Q2 & W2)  Practical boundaries of the monotonicity
>
> **When is it valid?**
>
> Our method is specifically designed for unknown Dynamic Range Compression (uDRC). This includes tasks where the degradation is functionally agnostic but physically monotonic, such as low-light enhancement, HDR reconstruction, and field-dependent contrast shifts in MRI. The framework's generality extends to any 1D or 2D signal suffering from unknown amplitude compression (e.g., Audio Declipping).
>
> **When does it fail?**
> 1. **Extreme Sensor Anomalies**: In certain extreme cases, severe overexposure causes charge overflow or inversion, leading the brightest light sources to appear black in the image (e.g., solarization). This represents a typical non-monotonic degradation.
> 2. **Geometric Mixing**: Degradations involving spatial scattering (e.g., severe blur, heavy fog, or underwater scattering) mix signals from neighboring pixels. In such scenarios, the local intensity order is altered by adjacent light sources, meaning a monotonic mapping is insufficient to describe these complex degradations.
>
> Meanwhile, we have summarized the above discussions and added them to the original manuscript.
>
> # (W3) Regarding our novelty
>
> Our contribution lies in the rigorous translation of a universal physical law into a neural architecture—Cascaded Bernstein Polynomials—rather than mere "engineering." By embedding monotonicity as a hard architectural constraint, we resolve the bias-variance dilemma in operator estimation, ensuring high fitting capacity for UDRC with minimal parameters.
>
> We believe the framework's simplicity is its primary strength. Instead of complex, task-specific modules for low-light, HDR, or MRI, we abstracted their shared physical essence to solve these complex ill-posed problems within a unified, physically-grounded framework.

---

> > ### Author Rebuttal · Reviewer_DB5k · 2026-04-03
> >
> > My concerns have been addressed. The rebuttal provided clear explanations and clarifications, which I appreciate. I have increased my score accordingly.

---

> > > ### Author Response · Authors · 2026-04-03
> > >
> > > We sincerely thank you for your time and effort in reviewing our rebuttal. We are pleased that our clarifications resolved your concerns. Your insightful feedback has been invaluable in improving our work.

---

### Official Review · Reviewer_XTzj · 2026-03-09

**Soundness:** 3
**Presentation:** 3
**Significance:** 3
**Originality:** 3
**Overall Recommendation:** 5
**Confidence:** 5

**Summary:**

This paper addresses blind inverse problems under unknown dynamic range compression, unifying tasks like low-light enhancement and HDR reconstruction. Recognizing monotonicity as a shared physical invariant, the authors propose the cascaded monotonic Bernstein (CaMB) operator. Integrated with a plug-and-play diffusion framework, CaMB enforces strict monotonicity, enabling robust, zero-shot restoration of radiometric fidelity without paired training data.

**Compliance With Llm Reviewing Policy:**

Affirmed.

**Final Justification:**

The author has responded to my concerns point to point. I appreciate their efforts for experimental validation.

**Key Questions For Authors:**

The paper presents an interesting idea, but several aspects require further clarification before the claims can be fully supported. In particular, the current experimental evaluation appears limited, and some results do not fully align with the stated claims. There are also questions regarding the consistency between the theoretical assumptions and the experimental setup, as well as the actual role of the CaMB operator relative to the diffusion prior in challenging cases such as saturated regions. Clarifying these points in the rebuttal would help better assess the validity and robustness of the proposed approach.

**Limitations:**

See weakness.

**Strengths And Weaknesses:**

Paper Strengths
1.	The paper formulates a unified perspective for unknown dynamic range compression across multiple tasks, including low-light enhancement, HDR reconstruction, and MRI recovery. This cross-domain formulation provides an interesting conceptual framework.
2.	The proposed CaMB operator introduces a structured monotonic function parameterization for modeling unknown compression curves. This design is simple, interpretable, and incorporates physical monotonicity constraints into the forward operator estimation.

Paper Weaknesses
Experiments
1.	The core experiments, particularly those related to HDR reconstruction, primarily rely on synthetically degraded images derived from ImageNet. However, commonly used real-world HDR or multi-exposure datasets are not included in the evaluation. Since synthetic degradations may not fully capture the complex processing introduced by real camera ISP pipelines, the current experimental setup may not adequately reflect real-world imaging scenarios. It would strengthen the paper if the authors could additionally evaluate the proposed method on real captured HDR or multi-exposure datasets.
2.	Although presented as a zero-shot method, the paper compares only against outdated baselines from 2023 and earlier. It neither introduces the latest fully-supervised SOTA as an "oracle upper bound" to quantify the performance trade-off, nor provides validation on out-of-distribution (OOD) scenarios, failing to justify the high inference cost during testing for practical applications.
3.	Despite the claim of producing "accurate colors," the qualitative results in Figure 3 suggest noticeable color drifting in some cases, where CaMB-Diff visually underperforms AGLLDiff (e.g., LOL v2 Real). This observation is also reflected in Table 1, where CaMB-Diff reports lower SSIM scores. In addition, the HDR evaluation includes only two examples, which is insufficient to demonstrate robustness under complex lighting conditions.

Methodology
4.	The paper models UDRC as a pixel-wise mapping without spatial mixing (Assumption 3.1). However, the low-field MRI experiment in the appendix introduces a 1D Gaussian blur kernel, which corresponds to spatial convolution degradation. This appears inconsistent with the stated modeling assumption. It would be helpful for the authors to clarify how the proposed CaMB operator handles such spatial degradations and whether this setting aligns with the theoretical framework.
5.	The paper attributes the HDR reconstruction capability largely to the proposed CaMB operator. However, in truly saturated regions (hard clipping), the camera response becomes flat (zero derivative), making the mapping mathematically non-invertible. In such cases, recovering detailed textures is likely to rely primarily on the diffusion prior rather than the physical constraint imposed by CaMB. This raises a question about the actual role of CaMB in saturated regions. Additional analysis or ablation studies would help clarify the respective contributions of the physical operator and the diffusion prior.

---

> ### Author Rebuttal · Authors · 2026-03-30
>
> We sincerely thank you for the constructive comments. Detailed responses follow below.
>
> ---
>
> # (Q1) Real HDR Evaluation
> We evaluate our method and zero-shot baselines on the real single-frame HDR dataset HDR-Eye [1]. Since our diffusion model is trained on 8-bit ImageNet, our goal is to recover an enhanced image with normal exposure. **Table R3** reports the quantitative results, and **Fig. R3** (https://anonymous.4open.science/r/CaMB-Diff/) provides visualizations, demonstrating that our approach successfully restores hidden details in extreme shadows while refining textures in brighter regions without overexposure.
>
> ||PSNR|SSIM|LPIPS|
> |:-|:-:|:-:|:-:|
> |GDP|15.35|0.695|0.191|
> |TAO|14.82|0.642|0.371|
> |**Ours**|**17.78**|**0.788**|**0.188**|
>
> *Table R3: Quantitative comparisons on real HDR-Eye [1].*
> > [1] Nemoto, Hiromi, et al. "Visual attention in LDR and HDR images." VPQM 2015.
>
> # (Q2) Comparisons of Latest SOTA (HVI)
> We added a comparison with the latest supervised SOTA, **HVI (CVPR 2025)** [2], on the low-light image enhancement (LLIE) task.
> + **In-domain**:  Quantitative and qualitative results are shown in **Table R4** and **Fig R4** .
> + **OOD**: To validate the OOD performance of supervised methods, we evaluated models trained on LOLv2-syn on the LOLv1 dataset, shown in **Table R5** & **Fig. R5** (https://anonymous.4open.science/r/CaMB-Diff).  Supervised methods exhibit severe performance degradation due to domain shifts, whereas CaMB-Diff outperforms HVI by **6.5 dB in PSNR**.
>
> ||Dataset|PSNR|SSIM|LPIPS|
> |:-|:-|:-:|:-:|:-:|
> |HVI|LOLv1|24.19|0.940|0.114|
> |HVI|LOLv2-R|24.72|0.941|0.131|
> |HVI|LOLv2-S|26.17|0.968|0.054|
>
> *Table R4: Quantitative results of HVI on three LLIE datasets.*
> ||PSNR|SSIM|LPI|FID|LOE|
> |-|-|-|-|--|-|
> |Retinx.|11.54|0.604|0.283|95.53|199.95|
> |HVI|13.63|0.727|0.315|108.08|185.14|
> |**Ours**|**20.14**|**0.849**|**0.217**|**72.65**|**82.21**|
>
> *Table R5: Quantitative comparison of OOD performance of HIV and Retinexformer.*
>
> > [2] Yan, Qingsen, et al. "Hvi: A new color space for low-light image enhancement." CVPR 2025.
>
> # (Q3) Color Distoration & Complex Lighting Conditions Evaluation
>
> We observe color distortion (**primarily darker tones than the reference**) in extremely low-light regions. This issue arises in severely underexposed areas, where the restoration is highly sensitive to the balance between the diffusion prior and the data fidelity term.
>
> This balance is controlled by the guidance scale $\lambda_t$. Specifically, we adopt the schedule $\lambda_t = \bar{\alpha}_t / (1 - \bar{\alpha}_t)$, where $\bar{\alpha}_t$ follows the diffusion noise schedule in the original manuscript. Under this setting, the data fidelity term dominates in the early stage, biasing the reconstruction toward low-intensity observations and leading to darker outputs.
>
> To alleviate this issue, we propose a new U-shaped $\lambda_t$ schedule that strengthens the diffusion prior in the early stage:
> $\lambda_t = \bar{\alpha}_t / (1 - \bar{\alpha}_t) + C(t/T)^k$.
> This improves the balance between prior and data consistency, resulting in more faithful color reconstruction.
>
> Results are shown in **Table R6** and **Fig R6** (https://anonymous.4open.science/r/CaMB-Diff).
> |   | PSNR | SSIM|
> |:-|:-|:-:|
> | Original $\lambda_t$ | 18.35 | 0.8728  |
> | U-shape $\lambda_t$ | 19.08 | 0.8748  |
>
> *Table R6: Quantitative analysis  of  $\lambda_t$ scheduling strategies on LOLv2 synthetic dataset.*
>
>
> # (Q4) Low-field (LF) MRI Degradation
> Compared with high-field MRI images, LF images exhibit not only **intensity discrepancies** but also **spatial resolution degradation**.
> Therefore, we decompose the LF degradation process into a **known spatial degradation** and an **unknown UDRC degradation**. Leveraging the specific acquisition protocols, the spatial degradation is accurately represented by a pre-defined Gaussian kernel. Crucially, incorporating this known spatial prior does not conflict with—or invalidate—our assumption regarding the UDRC forward operator (**Assumption 3.1**), as they operate in orthogonal domains (spatial vs. intensity). We provide further theoretical justification and empirical validation for Assumption 3.1 in our detailed response to `Reviewer muF6 (Q2)`.
>
>
> # (Q5) Respective Contributions of the Physical Operator and the Diffusion Prior
> We highlight that in authentic saturated (hard-clipped) regions, measurements provide no informative gradient. Thus, restoration depends entirely on the diffusion model's ability to generate textures from global context. CaMB’s structural monotonicity allows it to "lock" these regions with zero DC gradients, effectively delegating the restoration to the generative prior. Other blind methods (e.g., TAO, GDP) lack this structural constraint; their optimization often yields non-zero, erroneous gradients in clipped regions. These "noisy" gradients conflict with the diffusion prior, effectively penalizing the synthesized textures and leading to failure in saturated regions.

---

> > ### Author Rebuttal · Reviewer_XTzj · 2026-04-02
> >
> > The author has responded to my concerns point to point. I appreciate their efforts for experimental validation.

---

> > > ### Author Response · Authors · 2026-04-02
> > >
> > > We sincerely thank you for the time and effort you dedicated to reviewing our rebuttal. We are encouraged to hear that our additional experiments effectively resolved your concerns. Your insightful suggestions were invaluable in enhancing the quality of our work.

---

### Official Review · Reviewer_muF6 · 2026-03-11

**Soundness:** 4
**Presentation:** 4
**Significance:** 3
**Originality:** 3
**Overall Recommendation:** 5
**Confidence:** 4

**Summary:**

This paper studies a blind image restoration problem called unknown dynamic range compression (UDRC). In this problem, the original signal is changed by an unknown monotone function before it is observed. The authors propose CaMB-Diff, a method that uses a Bernstein polynomial model to describe this unknown degradation, and a diffusion model to recover the image. Experiments on low-light enhancement, HDR reconstruction, and low-field MRI enhancement show that the method produces better results and gives a better trade-off between flexibility and stability than previous zero-shot methods.

**Compliance With Llm Reviewing Policy:**

Affirmed.

**Final Justification:**

Thanks to the authors for providing additional experimental details. I will maintain my rating.

**Key Questions For Authors:**

1. Could the authors provide more theoretical support and validation for Assumption 3.1?
2. Could more experiments be provided (related to "Weakness" section of the review) ?

**Limitations:**

Yes, the authors have adequately discussed the limitations and potential negative societal impact of their work.

**Strengths And Weaknesses:**

Strengths:
1. The paper studies a blind inverse problem under unknown dynamic range compression. This problem is challenging and also has practical value.
2. The authors combine monotonic Bernstein polynomials with a diffusion prior. This idea is quite natural and fits the structure of the problem well.
3. The method has a relatively broad scope. It is not limited to one single task, and is tested on low-light enhancement, HDR reconstruction, and low-field MRI enhancement, which shows a certain level of generality.
4. The overall motivation of the paper is clear, and the connection between the method design and the experiments is also fairly clear.

Weakness:
More analysis experiments are expected:
1. More ablation studies of CaMB in different domains.
2. More analysis and experiments on the choice of hyper-parameters.

---

> ### Author Rebuttal · Authors · 2026-03-30
>
> Thank you for taking the time to review our work. We are pleased to receive your positive feedback. Below, we provide point-to-point responses to address your concerns.
>
> ---
>
> # (Q1) Validation for Assumption 3.1
> We conduct an empirical analysis of Assumption 3.1.
>
> To validate that the response value depends strictly on the signal intensity in Assumption 3.1 , we extract paired image data, where X denotes the normal-light image and Y represents the illumination-degraded image. We map all co-located pixel pairs $(X_{i, j},Y_{ i, j})$ from the entire image onto a 2D coordinate system, where the horizontal and vertical axes denote the pixel intensities of the normal-light image and the degraded image, respectively. If Assumption 3.1 holds, these pixel pairs should distribute closely around a monotonically increasing curve, rather than being randomly scattered across the entire 2D space. As illustrated in **Fig. R2(a)**(https://anonymous.4open.science/r/CaMB-Diff/), our statistical analysis of the pixel pair distributions across various datasets reveals that these points densely cluster along a single monotonic curve.
>
> To validate that the spatial independence of the signal response in Assumption 3.1, we sample three local patches with distinct textures (e.g., sky, building, and grass) from the selected images. Using isotonic regression, we independently fit three local curves for the pixel pairs within these respective patches and compare them with the global curve fitted on the entire image. As depicted in **Fig. R2(b)**(https://anonymous.4open.science/r/CaMB-Diff/), these four curves exhibit remarkable consistency. This strongly indicates that regardless of variations in local textures, the underlying radiometric degradation experienced by the pixels remains identical. We will include this study in the revised paper.
>
>
> # (Q2 & W1, 2) More Comprehensive Evaluation & Ablation Studies
>
> **1.Experiments in more complex scenarios.**
>
> To validate the effectiveness of our algorithm in handling real-world camera ISP effects and its robustness under complex illumination, we further evaluated our method on diverse real-world HDR scenarios. These include intricate indoor macro shots and wide-angle outdoor landscapes, as demonstrated in **Table R3**('`See Q1 of Reviewer XTzj`) and **Fig. R3** (https://anonymous.4open.science/r/CaMB-Diff) shows the visualization results.
>
> **2.Choice of guidance scale $\lambda_t$ schedule.**
>
> To validate the effectiveness of our proposed U-shaped $\lambda_t = \bar{\alpha}_t / (1 - \bar{\alpha}_t) + C(t/T)^k$ schedule, we compared it against the original $\lambda_t = \bar{\alpha}_t / (1 - \bar{\alpha}_t)$ schedule derived from the standard HQS decomposition. As shown in **Table R6**('`See Q3 of Reviewer XTzj`), the U-shaped schedule achieves a significant PSNR gain of 0.73 dB. Furthermore, qualitative results in **Fig. R6** (https://anonymous.4open.science/r/CaMB-Diff) demonstrate that the U-shaped strategy yields superior color fidelity that more closely aligns with the Ground Truth.
>
> **3.Impact of Diffusion Sampling Steps T.**
>
> To evaluate the influence of the reverse sampling trajectory on both generative quality and operator estimation, we conducted an ablation study by varying the total sampling steps $T \in  [20, 50, 100, 200, 500 ]$ on the LOLv1 dataset.  As shown in **Table R7**,  T=100 achieves an optimal balance between performance and computational efficiency.
>
> |Steps|PSNR|SSIM|
> |---|:-:|:-:|
> |20|16.04|0.7601|
> |50|18.67|0.8149|
> |**100**|**20.14**|0.8498|
> |200|20.08|**0.8510**|
> |500|19.98|0.8476|
>
> *Table R7:  Quantitative analysis of diffusion sampling steps T on the LOLv1 dataset.*
>
> **4. Choice of CaMB Depth(K) and Width(N)**
>
> The selection of $N$ and $K$ involves a fundamental trade-off between expressivity and optimization stability, where we prioritize increasing depth ($K \ge 6$) over width to ensure sufficient representational capacity. As $N=2$ may suffer from color drift (see **Fig. 6**), we advocate for $N=3$ or $4$ to strike an optimal balance for accurate radiometric restoration across diverse UDRC tasks.
> |CaMB Config|PSNR|SSIM|
> |---|:-:|:-:|
> |N=2 K=8|19.94|0.8476|
> |N=3 K=6|18.68| 0.8254|
> |**N=3 K=8**|**20.14**|**0.8498**|
> |N=3 K=12|20.04| 0.8416|
> |N=4 K=4| 16.31| 0.7446|
> |N=4 K=8|19.65|0.8267|
>
> *Table R8:  Ablation on CaMB Depth(K) and Degree (N) on the LOLv1 dataset.*

---

> > ### Author Rebuttal · Reviewer_muF6 · 2026-04-01
> >
> > Although the paper still has room for improvement, the authors’ response and the additional experimental results they provided have alleviated my concerns on this point. I am not especially confident but inclined to maintain my score.

---

> > > ### Author Response · Authors · 2026-04-02
> > >
> > > We sincerely thank you for your time and effort in reviewing our rebuttal. We are glad that our additional experiments alleviated your concerns. We will carefully incorporate your suggestions into our final version to further improve the manuscript’s quality and clarity.

---

### Official Review · Reviewer_9vC1 · 2026-03-13

**Soundness:** 3
**Presentation:** 2
**Significance:** 3
**Originality:** 3
**Overall Recommendation:** 4
**Confidence:** 4

**Summary:**

The paper is theoretically well-founded, providing proofs for the monotonicity of the atomic MBP layer and the universal approximation capability of the cascaded design. It also offers an original perspective by formulating UDRC as a structured manifold optimization problem and creatively employing Bernstein polynomials to enforce physical constraints instead of relying on soft regularization. By representing the forward operator with a compact and physically constrained parameterization, the method effectively alleviates the bias: variance trade-off in zero-shot blind restoration and demonstrates strong practical significance.

**Compliance With Llm Reviewing Policy:**

Affirmed.

**Final Justification:**

I appreciate the authors’ detailed response and have updated my score accordingly.

**Key Questions For Authors:**

Can you provide qualitative and quantitative comparisons with DIIP and DreamClean on the low-light enhancement task?

How do you explain the color distortion in Figure 3 and ground artifacts in Figure 5? Will you provide technical fixes or additional analysis to address these issues ?

Could you supplement runtime efficiency comparisons with existing zero-shot baselines (e.g., GDP, TAO, DIIP) to validate the practicality of CaMB-Diff?
Can you clarify the key differences between CaMB-Diff and DiffPIR (Zhu et al., 2023) in terms of optimization framework, operator modeling, and application scope?

**Limitations:**

No. The authors have not adequately discussed the limitations of CaMB-Diff (e.g., sensitivity to cascaded hyperparameters, performance degradation in extreme UDRC cases) or potential societal impacts. We suggest supplementing a dedicated "Limitations" section to address these gaps.

**Strengths And Weaknesses:**

Strengths and  weakness:

.1、This paper unifies diverse UDRC tasks by identifying monotonicity as a universal physical invariant across imaging modalities.

2、This paper designs the CaMB operator to resolve the bias-variance dilemma with inherent physical consistency and optimization stability.

3、This paper fuses the CaMB operator with diffusion priors for disentangled radiometric correction and detail recovery, achieving SOTA zero-shot UDRC performance.

Weakness:
1、	Qualitative results reveal non-negligible artifacts (e.g., color distortion in Figure3, ground artifacts in Figure 5) compared to ground truth, indicating limitations in radiometric fidelity. Additionally, no runtime efficiency comparisons are provided, despite potential concerns about the cascaded optimization’s time cost.
2、The statement that “standard solvers typically rely on a known forward operator or predefined analytical form” appears somewhat overstated. Recent concurrent works (e.g., DIIP and DreamClean) have demonstrated effective blind restoration without explicitly modeling the degradation operator. These relevant developments are not acknowledged or discussed.
3、Key state-of-the-art diffusion-based fully blind restoration methods (DIIP, DreamClean) are missing from the experimental comparisons, undermining the validity of the claimed SOTA performance in zero-shot UDRC.
4、The manuscript occasionally contains awkward phrasing (e.g., the repeated use of the vague term “physical anchor”) and inconsistent quotation mark usage, which affects readability. In addition, the relationship with closely related work such as DiffPIR is only briefly cited but not clearly differentiated, making the novelty boundary less explicit.
5、While the integration of monotonic constraints with diffusion models is technically interesting, the paper does not clearly explain how this design advances beyond existing plug-and-play diffusion frameworks (e.g., DiffPIR) or prior monotonic operator constructions.

---

> ### Author Rebuttal · Authors · 2026-03-30
>
> Thank you for the thoughtful review and constructive suggestions. We have addressed the identified weaknesses and provided detailed responses below.
>
> ---
>
> # (Q1 & W2, 3) Compared with DIIP & DreamClean
> We added comparisons of DIIP [1] and DreamClean [2] on the low-light image enhancement (LLIE) task using identical diffusion weights, consistent with the proposed CaMB-Diff. **Table R1** and **Fig. R1** (https://anonymous.4open.science/r/CaMB-Diff/) shows the quantitative and qualitative results, indicating that both methods fail to restore accurate exposure.
>
> We indicate that DIIP and DreamClean are effective on the **blind spatial degradation** tasks (e.g., deblur & SR), where the degraded image and ideal image have **compared intensities**. However, for LLIE (or URDC) task, there are **massive intensity gaps**.  Without specialized constraints, the standard data consistency term $\||g(z) - y||_2^2$ in DIIP forces the model to match the dark observation, inevitably pulling the reconstruction away from the natural image manifold and leaving it underexposed (Fig. R1)
>
> To demonstrate CaMB’s necessity, we integrated it into the DIIP framework by modifying the objective to: $\min_z ||\mathcal{M}_\Theta(g(z)) - y||_2^2$.  This allows CaMB to bridge the intensity gap, ensuring that the optimized $g(z)$ remains within the well-exposed range. Table R1 shows **DIIP+CaMB** yields a significant 11.8 dB PSNR gain over native DIIP, confirming that CaMB is essential for intensity-shifting tasks. We will include this study in the revised paper.
>
> |Method|PSNR|SSIM|LPIPS|
> |:--|:--:|:--:|:--:|
> |DIIP|7.37|0.156|0.737|
> |DreamClean|7.73|0.170|0.643|
> |**DIIP+CaMB**|19.16|0.829|0.310|
> |**Ours (CaMB-Diff)**|**20.14**|**0.850**|**0.217**|
>
> *Table R1: Quantitative Comparison on LLIE Task.*
>
> > [1] Chihaoui, H., & Favaro, P.  Diffusion Image Prior, ICCV. 2025
> >
> > [2] Xiao, J., et al. Dreamclean: Restoring clean image using deep diffusion prior. ICLR 2024
>
> # (Q2 & W1) Regarding Color Distoration & Ground Artifacts
>
> We clarified the color distortion issues and made more ablation studies (`See Q3 of Reviewer XTzj`).
>
> We observe artifacts in the snow-covered region in Fig. 5 of the HDR task, where the signal is severely truncated. In such regions, the reconstruction depends heavily on the diffusion prior to infer missing details. In the original manuscript, we incorporate multiple steps of unconditional diffusion sampling after each optimization iteration (similar to DreamClean-VPS) to enhance perceptual quality.  As shown in **Fig. R7**(https://anonymous.4open.science/r/CaMB-Diff/), while this improves visual richness, it can introduce inconsistencies due to an **imbalance between the diffusion prior and the data fidelity constraint**.
> In contrast, when we reduce or remove the reliance on diffusion sampling and emphasize data consistency, the reconstruction exhibits significantly improved fidelity to the reference image.
>
>
> # (Q3) Computational Efficiency Comparisons
> We compared the runtime and VRAM usage of zero-shot methods on a 256x256 RGB image (NVIDIA RTX 4090) fixing DIIP to 300 iterations for its best performance. As shown in **Table R2**, our method achieves the fastest optimization (20.4s) and the lowest VRAM usage. Our **CaMB design** reduces complexity from $O(n^k)$ (standard degree-$n^k$ polynomial) to **linear $O(nk)$** ($k$ degree-$n$ polynomials). This overhead is negligible compared to optimizing CNNs or affine matrices, enabling rapid inference.
>
> ||VRAM (GB)|Runtime (s)|
> |---|:-:|:-:|
> |DIIP|22.4|233.9|
> |GDP|2.9|168.7|
> |TAO|3.0|99.4|
> |**Ours**|**2.8**|**20.4**|
>
> *Table R2: Comparisons of zero-shot methods on VRAM usage and runtime.*
>
> # (Q3 & W4, 5) Key Differences between DiffPIR
> Thank you for the comments. We would clarify the major differences and revise the manuscript to emphasize our contribution.
> + **Scope**: DiffPIR is for the **non-blind** inverse problems with the known forward model $\boldsymbol{A}$, while CaMB-Diff addresses a kind of **blind inverse problem**, UDPC, where the forward model **lacks an explicit parse formulation**.
> + **Operator Modeling**: DiffPIR solves the non-blind inverse problems with a well-defined forward model, thus eliminating the need to model the forward model. CaMB-Diff models the unknown intensity compression with the proposed **monotonic Bernstein polynomials**
> + **Optimization framework**: Both DiffPIR and CaMB-Diff use the PnP-HQS framework to decouple the optimization objective into data-consistency and prior subproblems. While DiffPIR derives **closed-form solutions** with known linear operators.  In contrast, due to our unknown and non-linear operator, CaMB-Diff uses **gradient descent** to jointly estimate the unknown operator parameters and the latent image.

---

> > ### Author Rebuttal · Reviewer_9vC1 · 2026-04-02
> >
> > I appreciate the authors’ detailed response and have updated my score accordingly.

---

> > > ### Author Response · Authors · 2026-04-02
> > >
> > > We sincerely thank you for the time and effort you dedicated to reviewing our rebuttal. Your constructive suggestions were instrumental in improving the quality and clarity of our manuscript.

---

### Decision · Program_Chairs · 2026-04-30

**Decision:**

Accept (regular)

**Comment:**

A positive post-rebuttal consensus was reached on this submission. The rebuttal was viewed as thorough and constructive, with the reviewers’ concerns considered adequately addressed. Given the final reviewer consensus and the positive post-rebuttal assessments, the recommendation is to accept the paper.